# Neural surprise in somatosensory Bayesian learning

**Sam Gijsen** [1,4☯*], **Miro Grundei** [1,4☯*], **Robert T. Lange** [2,5], **Dirk Ostwald** [3], **Felix Blankenburg** [1]

**1** Neurocomputation and Neuroimaging Unit, Freie Universität Berlin, Germany, **2** Berlin Institute of Technology, Berlin, Germany, **3** Computational Cognitive Neuroscience, Freie Universität Berlin, Germany, **4** Humboldt-Universität zu Berlin, Faculty of Philosophy, Berlin School of Mind and Brain, Berlin, Germany, **5** Einstein Center for Neurosciences, Berlin, Germany

☯ These authors contributed equally to this work.
* sam.gijsen@fu-berlin.de (SG); m.grundei@fu-berlin.de (MG)

**Data Availability Statement:** The full, raw dataset can be found at: https://osf.io/83pgq/ with DOI 10.17605/OSF.IO/83PGQ The analysis and modeling code can be found at: https://github.com/SamGijsen/SurpriseInSomesthesis.

## Abstract

Tracking statistical regularities of the environment is important for shaping human behavior and perception. Evidence suggests that the brain learns environmental dependencies using Bayesian principles. However, much remains unknown about the employed algorithms, for somesthesis in particular. Here, we describe the cortical dynamics of the somatosensory learning system to investigate both the form of the generative model as well as its neural surprise signatures. Specifically, we recorded EEG data from 40 participants subjected to a somatosensory roving-stimulus paradigm and performed single-trial modeling across peristimulus time in both sensor and source space. Our Bayesian model selection procedure indicates that evoked potentials are best described by a non-hierarchical learning model that tracks transitions between observations using leaky integration. From around 70ms post-stimulus onset, secondary somatosensory cortices are found to represent confidence-corrected surprise as a measure of model inadequacy. Indications of Bayesian surprise encoding, reflecting model updating, are found in primary somatosensory cortex from around 140ms. This dissociation is compatible with the idea that early surprise signals may control subsequent model update rates. In sum, our findings support the hypothesis that early somatosensory processing reflects Bayesian perceptual learning and contribute to an understanding of its underlying mechanisms.

## Author summary

Our environment features statistical regularities, such as a drop of rain predicting imminent rainfall. Despite the importance for behavior and survival, much remains unknown about how these dependencies are learned, particularly for somatosensation. As surprise signalling about novel observations indicates a mismatch between one's beliefs and the world, it has been hypothesized that surprise computation plays an important role in perceptual learning. By analyzing EEG data from human participants receiving sequences of tactile stimulation, we compare different formulations of surprise and investigate the

**Funding:** This work was supported by Deutscher Akademischer Austauschdienst (SG, https://www.daad.de/en/), Humboldt-Universität zu Berlin, Faculty of Philosophy, Berlin School of Mind and Brain (SG & MG, http://www.mind-and-brain.de/home/), and Einstein Center for Neurosciences Berlin (RTL, https://www.ecn-berlin.de/). The funders had no role in study design, data collection and analysis, decision to publish, or preparation of the manuscript.

**Competing interests:** The authors have declared that no competing interests exist.

employed underlying learning model. Our results indicate that the brain estimates transitions between observations. Furthermore, we identified different signatures of surprise computation and thereby provide a dissociation of the neural correlates of belief inadequacy and belief updating. Specifically, early surprise responses from around 70ms were found to signal the need for changes to the model, with encoding of its subsequent updating occurring from around 140ms. These results provide insights into how somatosensory surprise signals may contribute to the learning of environmental statistics.

## Introduction

The world is governed by statistical regularities, such that a single drop of rain on the skin might predict further tactile sensations through imminent rainfall. The learning of such probabilistic dependencies facilitates adaptive behaviour and ultimately survival. Building on ideas tracing back to Helmholtz [1], it has been suggested that the brain employs an internal generative model of the environment which generates predictions of future sensory input. More recent accounts of perception and perceptual learning, including predictive coding [2, 3] and the free energy principle [4], propose that these models are continuously updated in light of new sensory evidence using Bayesian inference. Under such a view, the generative model is composed of a likelihood function of sensory input given external causes and a prior probability distribution over causes [4, 5]. Perception is interpreted as the computation of a posterior distribution over causes of sensory input and model parameters, while perceptual learning is seen as the updating of the prior distribution based on the computed posterior [6]. Such a description of Bayesian perceptual learning has been successfully used to explain aspects of learning in the auditory [7, 8, 9], visual [10, 11, 12], as well as somatosensory domain [13].

To investigate the underlying neuronal dynamics of perceptual inference, predictions formed by the brain can be probed by violating statistical regularities. Widely researched neurobiological markers of regularity violation include EEG components such as the auditory mismatch negativity (aMMN) and the P300 in response to deviant stimuli following regularity inducing standard stimuli. As an alternative to the oddball paradigm typically used to elicit such mismatch responses (MMR's) [14], the roving-stimulus paradigm features stimulus sequences that alternate between different trains of repeated identical stimuli [15]. Expectations are built up across a train of stimuli of variable length and are subsequently violated by alternating to a different stimulus train. The paradigm thereby allows for the study of MMR's based on the sequence history and independently of the physical stimulus properties. Analogues to the aMMN have also been reported for vision [16] and somatosensation (sMMN). The sMMN was first reported by Kekoni et al. [17] and has since been shown in response to deviant stimuli with different properties, including spatial location [18, 19, 20, 21, 22, 23, 24, 25, 26], vibrotactile frequency [17, 27, 28, 29], and stimulus duration [30, 31]. Increasing evidence has been reported for an account of the MMN as a reflection of Bayesian perceptual learning processes for the auditory [8, 32, 33], visual [12, 16], and to a lesser extent the somatosensory domain [13]. However, the precise mechanisms remain unknown, as it is unclear whether the MMN reflects the signaling of the inadequacy of the current beliefs or their adjustment, due to the lack of direct comparisons between these competing accounts.

In the context of probabilistic inference, the signalling of a mismatch between predicted and observed sensory input may be formally described using computational quantities of surprise [6, 34]. By adopting the vocabulary introduced by Faraji et al. [35] surprise can be grouped into two classes: puzzlement and enlightenment surprise. Puzzlement surprise refers

to the initial realization of a mismatch between the world and an internal model. Predictive surprise (PS) captures this concept based on the measure of information as introduced by Shannon [36]. Specifically, PS considers the belief about the probability of an event such that the occurrence of a rare event (i.e. an event estimated to have low probability of occurrence) is more informative and results in greater surprise. Confidence-corrected surprise (CS), as introduced by Faraji et al. [35] extends the concept of puzzlement surprise by additionally considering belief commitment. It quantifies the idea that surprise elicited by events depends on both the estimated probability of occurrence as well as the confidence in this estimate, with greater confidence leading to higher surprise. For example, in order for the percept of a drop of rain on the skin to be surprising, commitment to a belief about a clear sky may be necessary. The concept of enlightenment surprise, on the other hand, directly relates to the size of the update of the world model that may follow initial puzzlement. Bayesian surprise (BS) captures this notion by quantifying the degree to which an observer adapts their internal generative model in order to accommodate novel observations [37, 38].

Both predictive surprise [9] and Bayesian surprise [13] have been successfully applied to the full time-window of peri-stimulus EEG data to model neural surprise signals. However, the majority of studies have focused on P300 amplitudes, with applications of both predictive surprise [39, 40, 41, 42] and Bayesian surprise [40, 43, 44]. Earlier EEG signals have received less attention, although the MMN was reported to reflect PS [42]. Furthermore, due to the close relationship between model updating and prediction violation, only few studies have attempted to dissociate their signals. Although the use of different surprise functions in principle allows for a direct comparison of the computations potentially underlying EEG mismatch responses, such studies remain scarce. Previous research either focused on their spatial identification using fMRI [11, 45, 46, 47] or temporally specific, late EEG components [40]. Finally, to the best of our knowledge, only one recent pre-print study compared all three prominent surprise functions in a reanalysis of existing data, reporting PS to be better decoded across the entire post stimulus time-window [48].

Despite the successful account of perceptual learning using Bayesian approaches, the framework is broad and much remains unclear about the nature of MMR's, their description as surprise signals, and the underlying generative models that give rise to them. This is especially the case for the somatosensory modality, though evidence has been reported for the encoding of Bayesian surprise using the roving paradigm [13]. The current study expands on this work by recording EEG responses to a roving paradigm formulated as a generative model with discrete hidden states. We explore different mismatch responses, including the somatosensory analogue to the MMN, independent of the physical properties of stimuli. Using single-trial modeling, we systematically investigate the structure of the generative model employed by the brain. Having established the most likely probabilistic model, we provide a spatiotemporal description of its different surprise signatures in electrode and source space. As direct comparisons are scarce, we thus contribute by dissecting the dynamics of multiple aspects of Bayesian computation utilized for somatosensory learning across peri-stimulus time by incorporating them into one hierarchical analysis.

## Materials and methods

### Ethics statement

The study was approved by the local ethics committee of the Freie Universität Berlin (internal reference number: 51/2013) and written informed consent was obtained from all subjects prior to the experiment.

## Experimental design

**Participants.** 44 healthy volunteers (18-38 years old, mean age: 26, 28 females, all right-handed) participated for monetary compensation of 10 Euro per hour or an equivalent in course credit.

**Experimental procedure.** In order to study somatosensory mismatch responses and model them as single-trial surprise signals, we used a roving-stimulus paradigm [15]. Stimuli were applied in consecutive trains of alternating stimuli based on a probabilistic model (see below) with an inter-stimulus interval of 750ms (see Fig 1). Trains of stimuli consisted of two possible stimulation intensities. The first and last stimulus in a train were labeled as a deviant and standard, respectively. Thus, as opposed to a classic oddball design, the roving paradigm allows for both stimulus types to function as a standard or deviant.

Adhesive electrodes (GVB-geliMED GmbH, Bad Segeberg, Germany) were attached to the wrist through which the electrical stimuli with a 0.2ms duration were administered. In order to account for interpersonal differences in sensory thresholds, the two intensity levels were determined on a subject basis. The low intensity level (mean $5.05mA \pm 1.88$) was set in proximity to the detection threshold yet so that stimuli were clearly perceivable. The high intensity level (mean $7.16mA \pm 1.73$) was determined for each subject to be easily distinguishable from the low intensity level, yet remaining non-painful and below the motor threshold. The catch stimulus (described below) featured a threefold repetition of the 0.2ms stimulus at an interval of 50ms and was presented at either the low or high intensity level with equal probability.

Following familiarization with the electrical stimulation, 800 stimuli were administered in each of 5 experimental runs à 10 minutes. To ensure the subjects maintained attention on the electrical stimulation, they were instructed to count the number of catch trials (targets). In order to make the task non-trivial, the probability of the occurrence of a catch stimulus was set to either 0.01, 0.015, 0.02, 0.025, or 0.03, corresponding to a range of 3-32 trials per run. A subject received a stimulus sequence corresponding to each catch trial probability only once, with the order randomized between subjects. Following an experimental run, subjects indicated their counted number of catch trials and received feedback in the form of the correct amount.

**EEG data collection and preprocessing.** Data were collected using a 64-channel active electrode system (ActiveTwo, BioSemi, Amsterdam, Netherlands) at a sampling rate of 2048Hz, with head electrodes placed in accordance to the extended 10-20 system. Individual electrode positions were digitalized and recorded using an electrode positioning system (zebris Medical GmbH, Isny, Germany) with respect to three fiducial markers placed on the subject's face; left and right preauricular points and the nasion. This approach aided subsequent source reconstruction analyses.

Preprocessing was performed using SPM12 (Wellcome Trust Centre for Neuroimaging, Institute for Neurology, University College London, London, UK) and in-house scripts. First, the data were referenced against the average reference, high-pass filtered (0.01Hz), and down-sampled to 512Hz. Consequently, eye-blinks were corrected using a topological confound approach [49] and epoched using a peri-stimulus time interval of -100 to 600ms. All trials were then visually inspected and removed in case any significant artefacts were deemed to be present. The EEG data of four subjects were found to contain excessive noise due to hardware issues, resulting in their omission from further analyses and leaving 40 subjects. Finally, a low-pass filter was applied (45Hz). Grand mean somatosensory evoked potentials (SEPs) were calculated for deviant stimuli ('deviants') and for the standard stimuli directly preceding a deviant to balance the number of trials ('standards'). The preproccesed EEG data was baseline corrected with respect to the pre-stimulus interval of -100 to -5 ms. For the GLM analyses, each trial of the electrode data was subsequently linearly interpolated into a 32x32 plane for each

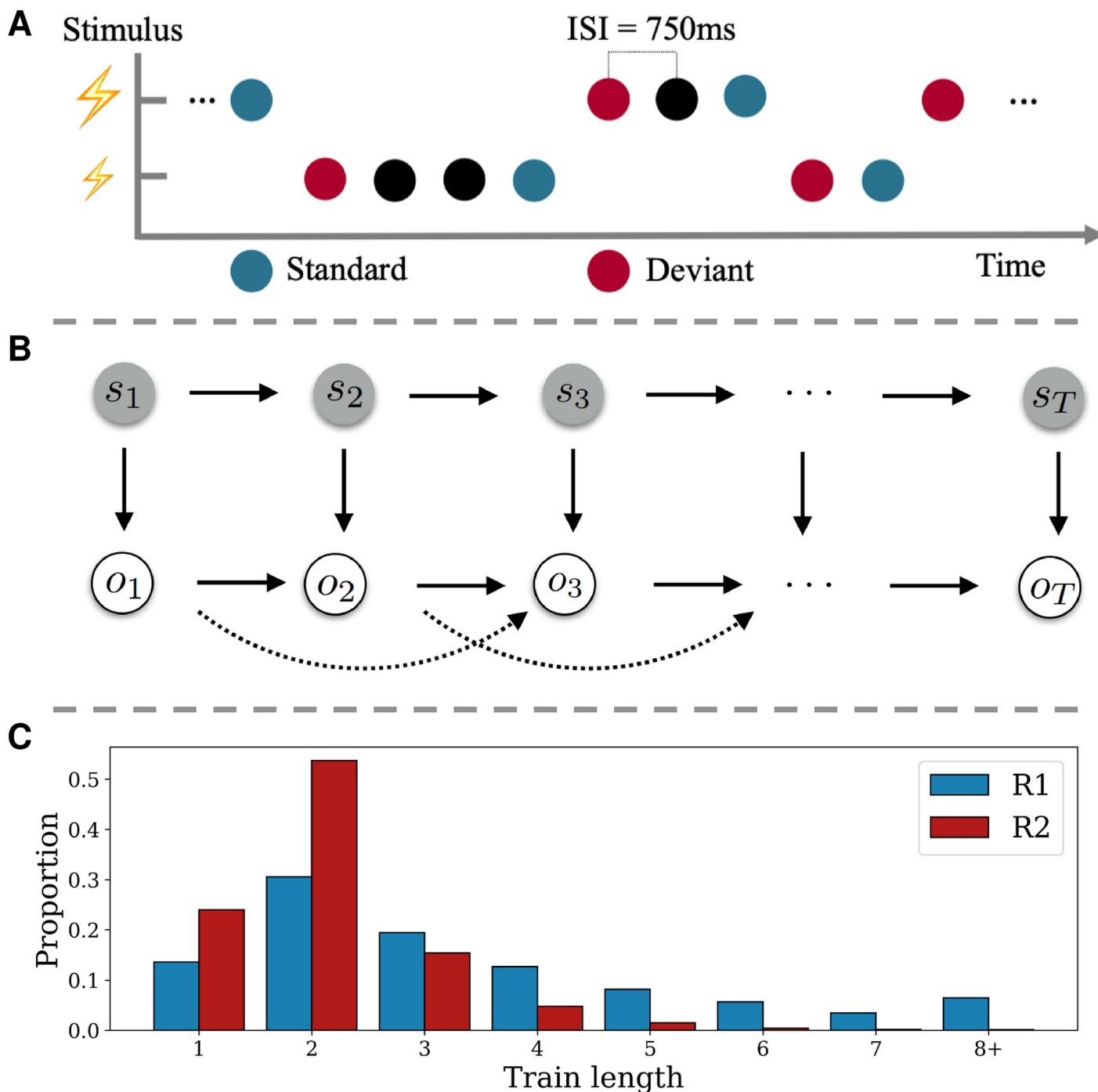

**Fig 1. Experimental design and stimulus generation.** A) Presentation of experimental stimuli using a roving-stimulus paradigm. Stimuli with two different intensities are presented. Their role as standard or deviant depends on their respective position within the presentation sequence. B) Graphical model of data-generating process. Upper row depicts the evolution of states $s_t$ over time according to a Markov chain. The states emit observations $o_t$ (lower row), which themselves feature second order dependencies on the observation level. C) Average proportion of resulting stimuli train lengths. Higher proportion of shorter trains for the fast switching regime ($R_2$; red) and more distributed proportion across higher train lengths for the slow switching regime ($R_1$; blue).

timepoint, resulting in a 32x32x308 image per trial. To allow for the use of random field theory to control for family-wise errors, the images were smoothed with a 12 by 12 mm full-width half-maximum (FWHM) Gaussian kernel. Catch trials were omitted for both the ERP and single-trial analyses.

## Generation of stimuli sequences

A property of generative models that is highly relevant for learning in dynamic environments is the manner by which they may adapt their estimated statistics in the face of environmental changes. By incorporating occasional switches between sets of sequence statistics, we aimed to compare generative models that embody different mechanisms of adapting to such change-points. Specifically, the sequential presentation of the stimuli originated from a partially observable probabilistic model for which the hidden state evolved according to a Markov chain (Fig 1) with 3 states $s$. The state transition ($p(s_t|s_{t-1})$) and emission probabilities $p(o_t|o_{t-1}, o_{t-2}, s_t)$ of the observations $o$ are listed in Table 1. One of the states was observable as it was guaranteed to emit a catch trial, while the other two states were latent, resembling fast and slow switching regimes. As the latter was specified with higher transition probabilities associated with repeating observations ($p(0|00)$ and $p(0|01)$) it thus produced longer stimulus trains on average. For every run, the sequence was initialized by starting either in the slow or fast switching regime with equal probability ($p(s_1) = \{0.5, 0.5, 0\}$, with catch probability being 0) and likewise producing a high or low stimulus with equal probability ($p(o_1|s_1) = \{0.5, 0.5\}$).

## Event-related potentials

To investigate the event-related response to the experimental conditions on the EEG data, the statistical design was implemented with the general linear model using SPM12. On the first level, the single-trial data of each participant was subject to a multiple regression approach with several regressors each coding for a level of an experimental variable: stimulus type (levels: standard and deviant), train length (levels: 2, 3, 4, 5, >6 stimuli) and a factor of experimental block as nuisance regressors (levels: block 1-5). An additional GLM with a balanced number of standard and deviant trials for the regimes (levels: fast and slow switching regime) showed no effect of regime or interaction of regime and stimulus type. The restricted maximum likelihood estimation implemented in SPM12 yielded $\beta$-parameter estimates for each model regressor over (scalp-)space and time which were further analysed at the group level. The second level consisted of a mass-univariate multiple regression analysis of the individual $\beta$ scalp-time images with a design matrix specifying regressors for stimulus type and regime as well as parametric regressors for train length and block and an additional subject factor. The

**Table 1. Data-generating process.**

| | State transition matrix | | | Sampling distribution |
|---|---|---|---|---|
| | $R_1$ | $R_2$ | $R_3$ | $p(o_t|o_{t-1}, o_{t-2}, s_t)$ |
| $R_1$ | $0.99 - \frac{1}{2}p(c)$ | $0.01 - \frac{1}{2}p(c)$ | $p(c)$ | $p(0|00) = 0.65, p(0|01) = 0.85, p(0|10) = 0.15, p(0|11) = 0.35$ |
| $R_2$ | $0.01 - \frac{1}{2}p(c)$ | $0.99 - \frac{1}{2}p(c)$ | $p(c)$ | $p(0|00) = 0.3, p(0|01) = 0.75, p(0|10) = 0.25, p(0|11) = 0.7$ |
| $R_3$ | $0.5 - \frac{1}{2}p(c)$ | $0.5 - \frac{1}{2}p(c)$ | $p(c)$ | $p(2) = 1$ |

Left: The state transition matrix. Right: Sampling distribution of the slow switching ($R_1$), fast switching ($R_2$), and catch-trial regime ($R_3$), emitting low intensity ($o_t = 0$), high intensity ($o_t = 1$), and catch stimuli ($o_t = 2$, with $p(c) = p(o_t = 2)$). Complementary probabilities are omitted (e.g. $p(1|00) = 1 - p(0|00)$).

condition contrasts were then computed by weighted summation of the group level regressors' $\beta$ estimates. To control for multiple comparisons, the scalp-time images were corrected with SPM's random field theory-based family wise error correction (FWE) [50]. The significant peaks of the GLM were further inspected by looking at their effect of train length and the corresponding $\beta$-parameter estimates of each train length were subjected to a linear fit for visualization purposes.

### Distributed source localization

In order to establish the somatosensory system as the driving dipolar generator of the EEG signals prior to 200ms, we followed a two-stage source reconstruction analysis consisting of a distributed and an equivalent current dipole (ECD) approach. While we report and model later EEG components in sensor-space, we refrained from source localizing these, as they most likely originate from a more distributed network of multiple sources [51, 52]. Furthermore, the somatosensory system has been shown to be involved in mismatch processing in the time window prior to 200ms [18, 19, 23, 26, 30, 53].

The distributed source reconstruction algorithm as implemented in SPM12 was used to determine the sources of the ERP's on a subject level. Specifically, subject-specific forward models were created using a 8196 vertex template cortical mesh which was co-registered with the electrode positions using the three aforementioned fiducial markers. SPM12's BEM EEG head model was used to construct the forward model's lead field. The multiple sparse priors under group constraints were implemented for the subject-specific source estimates [54, 55]. These were subsequently analyzed at the group level using one-sample t-tests. The yielded statistical parametric maps were thresholded at the peak level with $p < 0.05$ after FWE correction. The anatomical correspondence of the MNI coordinates of the cluster peaks were verified via cytoarchitectonic references using the SPM Anatomy toolbox. Details of the distributed source reconstruction can be reviewed in the results section.

### Equivalent current dipole fitting & source projection

The results of the distributed source reconstruction were subsequently used to fit ECDs to the grand average ERP data using the variational Bayes ECD fitting algorithm implemented in SPM12. The MNI coordinates resulting from the distributed source reconstruction served as informed location priors with variance of $10\text{mm}^2$ to optimize the location and orientation of the dipoles for a time-window around the peak of each component of interest (shown in the results section). For the primary somatosensory cortex (S1), two individual dipoles were fit to the time windows of the N20 and P50 components, respectively, to differentiate two sources of early somatosensory processing. Furthermore, a symmetrical dipolar source was fit to the peak of the N140 component of the evoked response with an informed prior around the secondary somatosensory cortex. Subsequently, the single trial EEG data of each subject was projected with the ECD lead fields onto the 4 sources using SPM12, which enabled model selection analyses in source-space.

### Trial-by-trial modeling of sensor- and source-space EEG data

**Sequential Bayesian learner models for categorical data.** To compare Bayesian learners in terms of their generative models and surprise signals, we specified various probabilistic models which generate the regressors ultimately fitted to the EEG data. Capitalizing on the occasional changes to the sequence statistics included in the experimental stimulus generating model, we assess two approaches to latent state inference. Specifically, a conjugate Dirichlet-Categorical (DC) model as well as a Hidden Markov Model (HMM) [56] were used for

modeling categorical data. The DC model is non-hierarchical and does not feature any explicit detection of the regime-switches. However, it is able to adapt its estimated statistics to account for sequence change-points by favoring recent observations over those in the past, akin to a progressive "forgetting" or leaky integration. The model assumes a real-valued, static hidden state $s_t$ that is shared across time for each observation emission.

In contrast, the HMM is a hierarchical model for which $s_t$ is a discrete variable and assumed to follow a first order Markov Chain, mimicking the data generation process. As such, it contains additional assumptions about the task structure, which allows for flexible adaptation following a regime-switch by performing inference over a set of discrete hidden states $K$ ($s_t \in \{1, \ldots, K\}$). The transition dynamics are given by the row-stochastic matrix $\mathbf{A} \in \mathbb{R}^{K \times K}$ with $a_{ij} \geq 0$ and $\sum_{j=1}^{K} a_{ij} = 1$:

$$p(s_t|s_{t-1}) = \mathbf{A} \Leftrightarrow p(s_t^j|s_{t-1}^i) = a_{ij} \text{ for } t = 1, \ldots, T. \tag{1}$$

Within our two model classes, we differentiate between four probabilistic models. Here, the aim is to investigate which sequence statistics are estimated by the generative model. In the case of Stimulus Probability (SP) inference, the model does not capture any Markov dependence: $o_t$ solely depends on $s_t$. Alternation Probability (AP) inference captures a limited form of first-order Markov dependency, by estimating the probability of the event of altering observations $d_t$ given the hidden state $s_t$ and the previous observation $o_{t-1}$, where $d_t = \mathbf{1}_{o_t \neq o_{t-1}}$ takes on the value 1 if the current observation $o_t$ differs from $o_{t-1}$. With Transition Probability (TP$_1$) inference, the model accounts for full first-order Markov dependence and estimates separate alternation probabilities depending on $o_{t-1}$ and $s_t$, i.e. $p(o_t|o_{t-1}, s_t)$. Finally, TP$_1$ inference may be extended (TP$_2$) to also depend on $o_{t-2}$, and by estimating $p(o_t|s_t, o_{t-1}, o_{t-2})$ it most closely resembles the structure underlying the data generation.

**Dirichlet-Categorical model.** The Dirichlet-Categorical model is a simple Bayesian observer that counts the observations of each unique type to determine its best guess of their probability (Eq 5). Its exponential forgetting parameter implements a gradual discounting of observations the further in the past they occurred (Eq 8). It is part of the Bayesian conjugate pairs and models the likelihood of the observations using the Categorical distribution with $\{1, \ldots, M\}$ different possible realizations per sample $y_t$. Given the probability vector $\mathbf{s} = \{s_1, \ldots, s_M\}$ defined on the $M - 1$ dimensional simplex $\mathcal{S}_{M-1}$ with $s_i > 0$ and $\sum_{j=1}^{M} s_j = 1$, the probability mass function of an event is given by

$$p(y_t = j|s_1, \ldots, s_M) = s_j \tag{2}$$

Furthermore, the prior distribution over the hidden state $\mathbf{s}$ is given by the Dirichlet distribution which is parametrized by the probability vector $\alpha = \{\alpha_1, \ldots, \alpha_M\}$:

$$p(s_1, \ldots, s_M|\alpha_1, \ldots \alpha_M) = \frac{\Gamma(\sum_{j=1}^{M} \alpha_j)}{\prod_{j=1}^{M} \Gamma(\alpha_j)} \prod_{j=1}^{M} s_j^{\alpha_j - 1}. \tag{3}$$

Hence, we have a Dirichlet prior with $s_1, \ldots, s_M \sim Dir(\alpha_1, \ldots, \alpha_M)$ and a Categorical likelihood with $y \sim Cat(s_1, \ldots, s_M)$. Given a sequence of observations $y_1, \ldots, y_t$ the model then combines the likelihood evidence with prior beliefs in order to refine posterior estimates over the latent variable space (derivations of enumerated formulas may be found in the

supplementary material S1 Appendix):

$$
\begin{aligned}
p(s_1, \ldots, s_M | y_1, \ldots, y_t) \quad &\propto p(s_1, \ldots, s_M | \alpha_1, \ldots, \alpha_M) \prod_{i=1}^{t} p(y_i | s_1, \ldots, s_M) \\
&= \prod_{j=1}^{M} s_j^{\alpha_j - 1 + \sum_{i=1}^{t} \mathbf{1}\{y_i = j\}}
\end{aligned}
\tag{4}
$$

Since the Dirichlet prior and Categorical likelihood pair follow the concept of conjugacy, given an initial $\alpha^0 = \{\alpha_1^0, \ldots, \alpha_M^0\}$ (set as a hyperparameter) the filtering distribution can be computed:

$$
p(\mathbf{s}_t | y_1, \ldots, y_t) = p(s_1, \ldots, s_M | y_1, \ldots, y_t) = Dir(\alpha^t) \quad \text{with} \quad \alpha_j^t = \alpha_j^0 + \sum_{i=1}^{t} \mathbf{1}\{y_i = j\}.
\tag{5}
$$

Likewise, one can easily obtain the posterior predictive distribution (needed to compute the predictive surprise readout) by integrating over the space of latent states:

$$
\begin{aligned}
p(y_t = x | y_1, \ldots, y_{t-1}) \quad &= \int p(y_t = x | s_1, \ldots, s_M) p(s_1, \ldots, s_M | y_1, \ldots, y_{t-1}) d\mathcal{S}_M \\
&= \frac{\alpha_x^t}{\sum_{j=1}^{M} \alpha_j^t}
\end{aligned}
\tag{6}
$$

We can evaluate the likelihood of a specific sequence of events which can be used to iteratively compute the posterior:

$$
p(y_1, \ldots, y_t) = p(y_1) \prod_{i=2}^{t} p(y_i | y_{1:i}) = \frac{1}{M} \prod_{i=2}^{t} \prod_{j=1}^{M} \frac{\alpha_j^i}{\sum_{k=1}^{M} \alpha_k^i}
\tag{7}
$$

For the evaluation of the posterior distributions, we differentiate between three inference types which track different statistics of the incoming sequence as described above (for a graphical model see Fig 2):

1. The stimulus probability (SP) model: $y_t = o_t$ for $t = 1, \ldots, T$

2. The alternation probability (AP) model: $y_t = d_t$ for $t = 2, \ldots, T$

3. The transition probability model (TP$_1$ & TP$_2$): $y_t = o_t$ for $t = 1, \ldots, T$ with a set of hidden parameters $\mathbf{s}_1^{(i)}$ for each transition from $o_{t-1} = i$ and $\mathbf{s}_2^{(j)}$ for each transition from $o_{t-2} = j$ respectively

Despite a static latent state representation, the DC model may account for hidden dynamics by incorporating an exponential memory-decay parameter $\tau \in [0, 1]$ which discounts observations the further in the past they occurred. Functioning as an exponential forgetting mechanism, it allows for the specification of different timescales of observation integration.

$$
p(\mathbf{s}_t | y_1, \ldots, y_t) = p(s_1, \ldots, s_M | y_1, \ldots, y_t) = Dir(\alpha^t)
$$

$$
\text{with} \quad \alpha_j^t = \alpha_j^0 + \sum_{i=1}^{t} e^{-\tau(t-i)} \mathbf{1}\{y_i = j\}.
\tag{8}
$$

**Hidden Markov model.** While the Dirichlet-Categorical model provides a simple yet expressive conjugate Bayesian model for which analytical posterior expressions exist, it is limited in the functionality of the latent state $s$ due to its interpretation as the discrete distribution

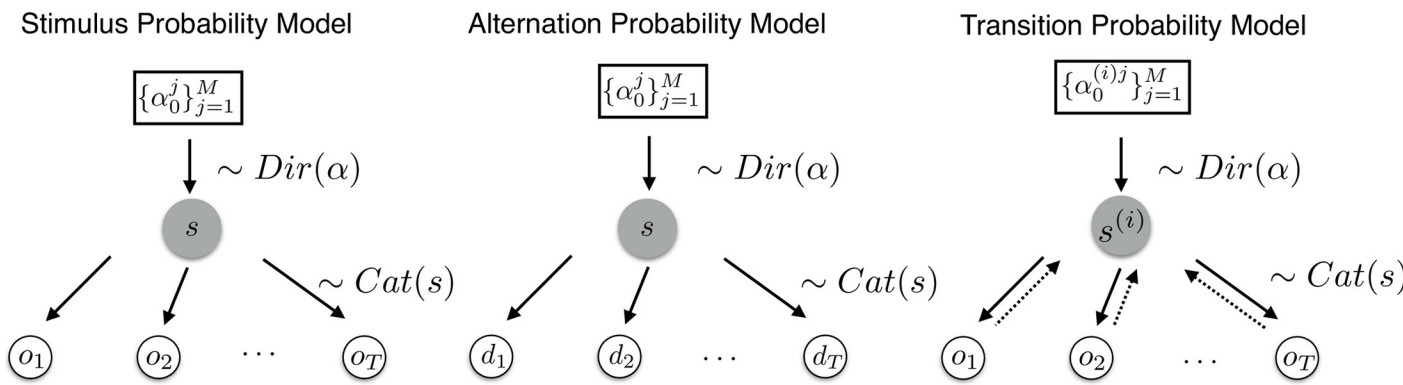

**Fig 2. Dirichlet-Categorical model as a graphical model.** Left: The stimulus probability model which tracks the hidden state vector determining the sampling process of the raw observations. Middle: The alternation probability model which infers the hidden state distribution based on alternations of the observations. Right: The transition probability model which assumes a different data-generating process based on the previous observations. Hence, it infers $M$ sets of probability vectors $\alpha^i$.

over categories. Hidden Markov Models (HMMs), on the other hand, are able to capture the dynamics of the hidden state with the transition probabilities of a Markov Chain (MC). Given the hidden state at time $t$, the categorical observation $o_t$ is sampled according to the stochastic matrix $\mathbf{B} \in \mathbb{R}^{M \times K}$, containing the emission probabilities, $p(o_t|s_t)$. The evolution of the discrete hidden state according to a MC, $p(s_t|s_{t-1})$, is described by the stochastic matrix $\mathbf{A} \in \mathbb{R}^{K \times K}$. The initial hidden state $p(s_1)$ is sampled according to the distribution vector $\pi \in \mathbb{R}^K$. $\mathbf{A}$, $\mathbf{B}$ are both row stochastic, hence $A_{ij}, B_{ij} \geq 0$, $\sum_{j=1}^{K} A_{ij} = 1$ and $\sum_{j=1}^{M} B_{ij} = 1$. The graphical model described by the HMM setup is thereby specified as depicted in Fig 3.

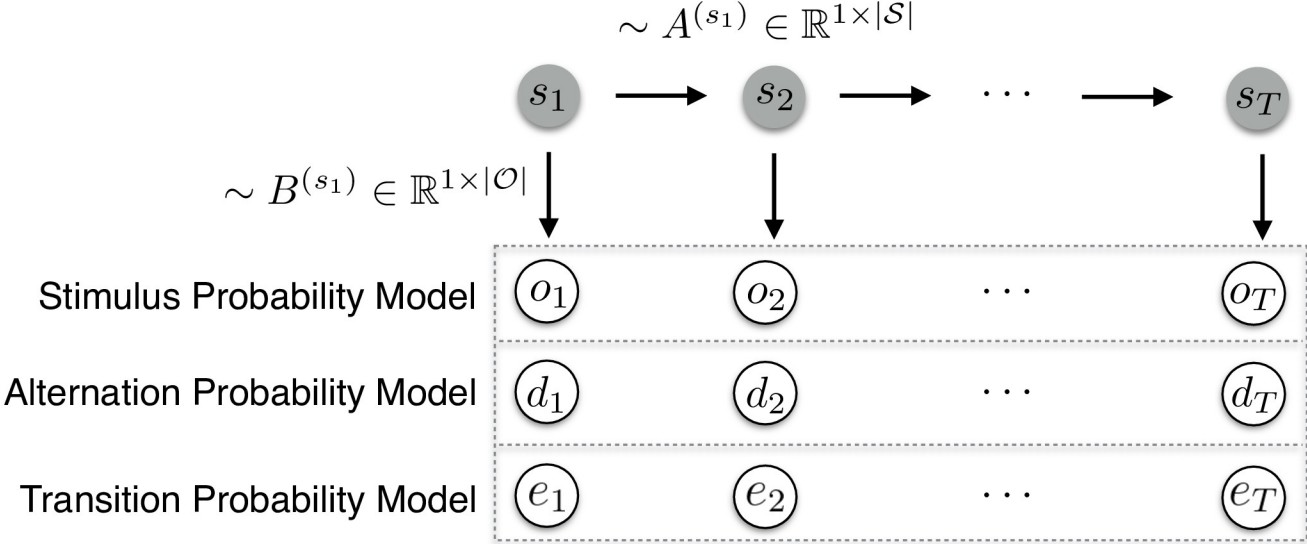

**Fig 3. Hidden Markov model as a graphical model.** Upper row depicts the evolution of states $s_t$ according to the transition matrix $A^{(s_t)}$. The states emit observational data (dotted rectangle) according to the probabilities specified in stochastic matrix $B^{(s_t)}$ which depends on the type of inference. The stimulus probability model infers the emission probabilities associated with the raw observations $o_t$. The alternation probability model tracks the alternations of observations with $d_t = \mathbf{1}_{o_t \neq o_{t-1}}$. The transition probability model assumes a data-generating process based on previous observations, with $e_t$ coding for the transitions between observations.

Classically, the parameters of this latent variable are inferred using the Expectation-Maximisation (EM) algorithm. Therefore, and in order to derive the factorisation of the joint likelihood $p(o_{1:t}, s_{1:t})$, the backward and forward probabilities are used in conjunction with the Baum-Welch algorithm in order to perform the inference procedure (see S1 Appendix).

**HMM Implementation.** The aim of the HMM was to approximate the data generation process more closely by using a model capable of learning the regimes over time and performing latent state inference at each timestep. To this end, prior knowledge was used in its specification by fixing the state transition matrix close to its true values ($p(s_t = s_{t-1}) = 0.99$). The rare catch trials were removed from the data prior to fitting the HMM and thus their accompanying third regime was omitted, resulting in a two-state HMM. Given that an HMM estimates emission probabilities of the form $p(o_t|s_t)$ and thus does not capture any additional explicit dependency on previous observations, the input vector of observations was transformed prior to fitting the models. For AP and TP inference this equated to re-coding the observation $o_t$ to reflect the specific event that occurred. Specifically, for the AP model the input sequence was $d_t = 1_{o_t \neq o_{t-1}}$, while for TP$_1$ and TP$_2$ a vector of events was used corresponding to the four possible transitions from $o_{t-1}$ or eight transitions from $o_{t-2}$ respectively. Thus, the HMM estimates two sets (reflecting the two latent states) of emission probabilities which correspond to these events ($y_t$). Despite this deviation of the fitted models from the underlying data generation process, the AP and TP models reliably captured R$_1$ and R$_2$ to their capability, with TP$_2$ retrieving the true, but unknown underlying emission probabilities (see S1 Fig). As expected, SP inference was agnostic to the regimes, while AP and TP inference allowed for the tracking of the latent state over time (S1 Fig). An example of the filtering posterior may be found in Fig 4.

**Surprise readouts.** For each of the probabilistic models described above, three different surprise functions were implemented, forming the predictors for the EEG data: predictive surprise $PS(y_t)$, Bayesian surprise $BS(y_t)$, and confidence-corrected surprise $CS(y_t)$. These may be

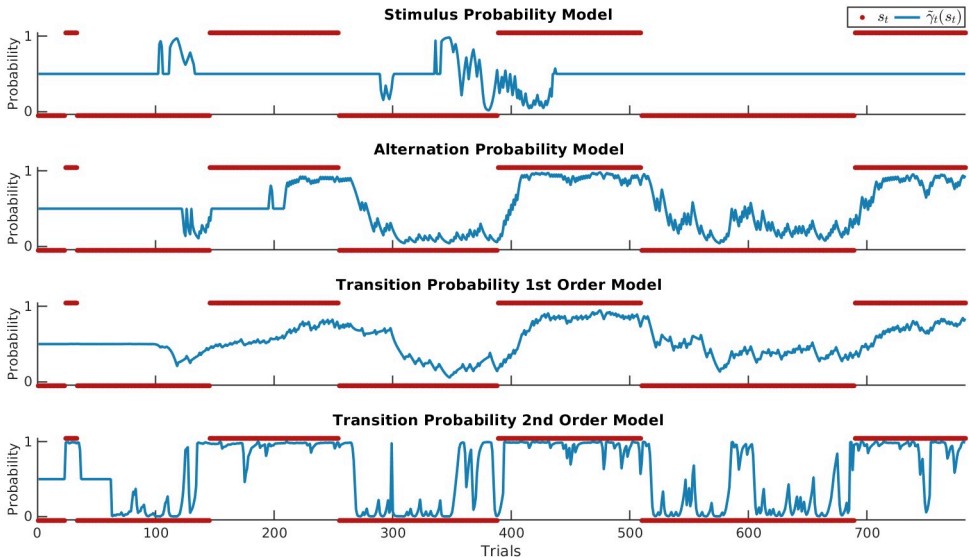

**Fig 4. Posterior probabilities of the HMM.** Comparison of the filtering posterior $\hat{\gamma}_t(s_t) = p(s_t|o_1, \ldots, o_t)$ of the different HMM inference models for an example sequence. The true, but unknown regimes of the data generation process are plotted in red. Note that, as the regimes were balanced in terms of stimulus probabilities, SP inference is not able to capture the underlying regimes and instead attempts to dissociate two states based on empirical differences in observed stimulus probabilities.

interpreted as read-out functions of the generative model, signalling a mismatch between the world and the internal model.

The predictive surprise is defined as the negative logarithm of the posterior predictive distribution $p(y_t|s_t)$:

$$PS(y_t) := -\ln p(y_t|s_t) = -\ln p(y_t|y_1, \ldots, y_{t-1}). \tag{9}$$

A posterior that assigns little probability to an event $y_t$ will cause high (unit-less) predictive surprise and as such is a measure of puzzlement surprise. The Bayesian surprise, on the other hand, quantifies enlightenment surprise and is defined as the Kullback-Leibler (KL) divergence between the posterior pre- and post-update:

$$BS(y_t) := KL(p(s_{t-1}|y_{t-1}, \ldots y_1) \| p(s_t|y_t, \ldots, y_1)) \tag{10}$$

Confidence-corrected surprise is an extended definition of puzzlement surprise which additionally considers the commitment of the generative model as it is scaled by the negative entropy of the prior distribution. It is defined as the KL divergence between the informed prior and posterior distribution of a naive observer, corresponding to an agent with a flat prior $\hat{p}(s_t)$ (i.e. all outcomes are equally likely) which observed $y_t$:

$$CS(y_t) := KL(p(s_t) \| \hat{p}(s_t|y_t)), \tag{11}$$

For the DC model, the flat prior $\hat{p}(s_t)$ can be written as $Dir(\alpha_1, \ldots, \alpha_m)$ with $\alpha_m = 1$ for $m = 1, \ldots, M$. The naive observer posterior $\hat{p}(s_t|y_t)$ simply updates the flat prior based on only the most recent observation $y_t$. Hence, we have $\hat{p}(s_t|y_t) = Dir(\alpha'_1, \ldots, \alpha'_m)$ with $\hat{\alpha}_m = 1 + \mathbf{1}_{y_t=m}$. A detailed account of the readout definitions can be found in S1 Appendix.

For the HMM, the surprise readouts are obtained by iteratively computing the posterior distribution via the Baum-Welch algorithm using the *hmmlearn* Python package [57]. For timestep $t$ this entails fitting the HMM for a stimulus sequence $o_1, \ldots, o_t$ which gives a set of parameter estimates, $\hat{\pi}_t, \hat{A}_t, \hat{B}_t$ and the filtering posterior $\hat{\gamma}_t(s_t) = p(s_t|o_1, \ldots, o_t)$. Predictive, Bayesian, and confidence-corrected surprise may then be expressed as follows (see S1 Appendix).

$$PS(o_{t+1}) \approx -\ln(\hat{B}_t^T \hat{A}_t^T \hat{\gamma}_t(s_t)) \tag{12}$$

$$BS(o_{t+1}) \approx \sum_k^K \hat{\gamma}_t(s_t = k) \ln \frac{\hat{\gamma}_t(s_t = k)}{\hat{\gamma}_{t+1}(s_{t+1} = k)} \tag{13}$$

Following Faraji et al. [35], confidence-corrected surprise may be expressed as a linear combination of predictive surprise, Bayesian surprise, a model commitment term (negative entropy) $C(p(s_t))$, and a data-dependent constant scaling the state space $O(t)$. Here we make use of this alternative expression of CS in order to facilitate the HMM implementation:

$$CS(o_t) = BS(o_t) + PS(o_t) + C(p(s_t)) + \ln O(t) \tag{14}$$

Fig 5 shows the regressors for an example sequence of the HMM TP$_1$ and DC TP$_1$ models with an observation half-life of 95. The PS regressors of both models show greater variability in the slow switching regime as compared to the fast-switching regime, where repetitions are more common (and consequently elicit less predictive surprise) while alterations are less common (and thus elicit greater surprise). As such, the PS regressors differ between regimes as a function of the estimated transition probabilities. The speed at which models adapt to the changed statistics depends on the forgetting parameter for the DC model while for the HMM it is

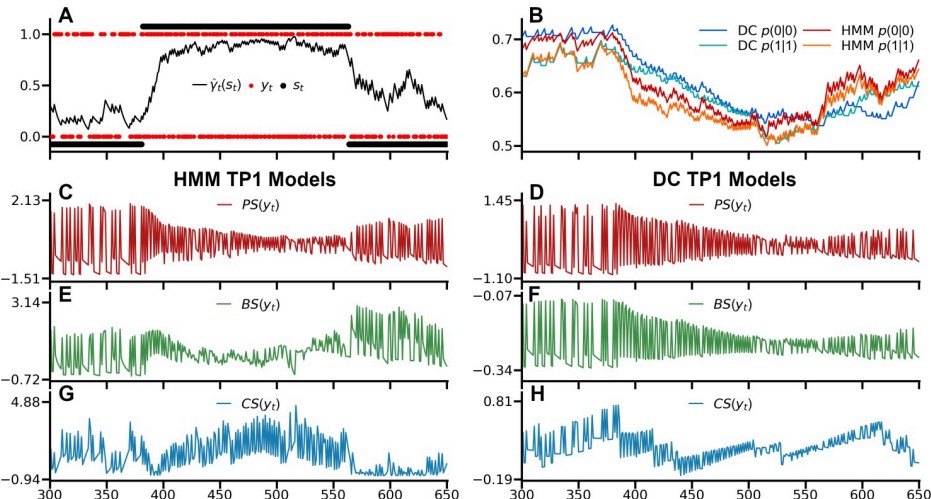

**Fig 5. Surprise readouts.** A) Example sequence with $o_t$ in red, $s_t$ in black with $s_t = 0$ for the slow-switching regime and $s_t = 1$ for the fast switching regime, and the HMM filtering posterior $\hat{\gamma}_t(s_t)$ in between. The rare catch-trials are not plotted to facilitate a direct comparison between the HMM and DC models. B) The normalized probability estimates of the HMM $TP_1$ and DC $TP_1$ model with an observation half-life of 95, displaying differences in estimates arising from different adaptations to regime switches. C,E,G) The z-scored surprise readouts of the HMM $TP_1$ models: predictive surprise (PS), Bayesian surprise (BS), and confidence-corrected surprise (CS). D,F,H) The z-scored surprise readouts of the DC $TP_1$ models.

dependent on the degree to which the regimes have been learned. BS is markedly distinct for the two models due to the differently modeled hidden state. DC BS features many small updates during the fast-switching regime, with more irregular, larger updates during the slow-switching regime, while HMM BS expresses the degree to which an observation produces changes in the latent state posterior. Finally, HMM CS is scaled by the confidence in the latent state posterior, tending to greater surprise the more committed the model is to one particular latent state, and lower surprise otherwise, such as at the end of the example sequence. Meanwhile, due to its static latent state, confidence for DC CS results only from commitment to beliefs about the estimated transition probabilities between observations themselves, with rare events causing drops in confidence. Taken together, the HMM regressors ultimately depend on its posterior over latent states, and while this is absent for the DC, its regressors display differences between the two regimes as a function of its integration timescale which in turn allows it to accommodate its probability estimates to the currently active regime.

In an exploratory analysis, the trial-definitions of the GLM analysis of the individual electrode-time point data were applied to the surprise readout regressors. This allowed for the derivation of model-based predictions for the observed beta-weight dynamics of the ERP GLM. First, we generated an additional 25000 sequences of 800 observations using the same generative model used for the subject-specific sequences. The averaged surprise readouts of these simulated sequences yielded model-derived predictions, which allowed for a visual verification of the presence of these predictions in the (200) experimental sequences. As each study subject was exposed to 5 sequences, these sequences were grouped into sets of 5 (yielding 5000 simulated subjects) to mirror the EEG analysis. Besides the HMM, we used the Dirichlet-Categorical models with different values for the forgetting-parameter ('no forgetting', long, medium-length and very short stimulus half-lives) (S2 Fig). To reduce the model-space, only $TP_1$ models were used for this analysis.

**Model fitting via free-form variational inference algorithm.** Each combination of model class (DC and HMM), inference type (SP, AP, TP$_1$, TP$_2$), and surprise readout function (PS, BS, CS) yields a stimulus sequence-specific regressor. The same models were used across subjects and as such the regressors did not include any subject specific parameters. These regressors, as well as those of a constant null-model, were fitted to the single-trial, event-related electrode and source activation data. Using a free-form variational inference algorithm for multiple linear regression [58, 59, 60], we obtained the model evidences allowing for Bayesian model selection procedures [61], which accounts for the accuracy-complexity trade-off in a formal and well-established manner [62]. In short, the single-subject, single peri-stimulus time bin data $y \in \mathbb{R}^{n \times 1}$ for $n \in N$ trials was modeled in the following form:

$$p(y, \beta, \lambda) = p(y|\beta, \lambda)p(\beta)p(\lambda) \tag{15}$$

with $\beta \in \mathbb{R}^p$ and $\lambda > 0$ denoting regression weights and observation noise precisions, respectively. The parameter-conditional distribution of $y$, $p(y|\beta, \lambda)$, is specified in terms of a multivariate Gaussian density with expectation parameter $X\beta$ and spherical covariance matrix. The design matrix $X$ consisted of a constant offset (null-model: $X \in \mathbb{R}^{n \times 1}$) and an additional surprise-model specific regressor in case of the non-null models ($X \in \mathbb{R}^{n \times 2}$). Both a detailed description of the algorithm and the test procedure performed on simulated data used to select the prior parameters for the variational distributions of $\beta$ and $\lambda$ may be found in the supplementary material S2 Appendix.

**Bayesian model selection.** Before modeling single subject, single peri-stimulus time bin data ($y$) as described above, the single-trial regressors of all non-null models as well as the data underwent z-score normalization to allow for the use of the same model estimation procedure for both sensor and source data. For single subjects, data and regressors corresponding to the five experimental runs were concatenated prior to fitting. To allow for the possibility that the brain estimates statistics computed across multiple timescales of integration [9, 63, 64], the forgetting-parameter $\tau$ of the DC model was optimized for each subject, model, and peri-stimulus time-bin. To this end, DC model regressors were fitted for a logarithmically spaced vector of 101 $\tau$-values on the interval of 0 to 1 and the value of $\tau$ that resulted in the highest model evidence was chosen. To penalize the DC model for having one of its parameters optimized, the degree to which $\tau$ optimization on average inflated model evidences was subtracted prior to the BMS procedure. Specifically, the difference in model evidence between its average for all parameter-values and the optimized value was computed and subsequently averaged across post-stimulus timebins, sensors, and subjects. It should be noted that the applied procedure constitutes a heuristic for the penalization of model complexity while no explicit parameter fitting procedure was implemented within model estimation.

The furnished model evidences were subsequently used for a random-effects analysis as implemented in SPM12 [61] to determine the models' relative performance in explaining the EEG data. In order to combat the phenomenon of model-dilution [65], a hierarchical approach to family model comparison was applied (for a graphical overview see S3 Fig). This amounts to a step-wise procedure that leads to data-reduction at subsequent levels. Note that this procedure is performed for each peri-stimulus time bin and electrode independently (resulting in 22976 model comparisons per subject). In a first step, the two model classes DC and HMM were compared against each other and the null-model in a family-wise BMS. A threshold of exceedance probabilities $\varphi > 0.99$ in favour of either the DC or HMM was applied, so that only whenever there was strong evidence in favour of one of the model classes over both the alternative and the null-model the following analyses were applied. As the current analyses are not statistical tests per se, the thresholding of the data by certain exceedance

probabilities ultimately constituted an arbitrary choice to reduce data in order to visualize (and draw conclusions on) effects with certain minimum probabilities within a large model space. For timepoints with exceedance probabilities above this threshold, a family-wise comparison of $TP_1$ and $TP_2$ was performed in order to determine which order of transition probabilities would be used for the second level. Subsequently, either the $TP_1$ or $TP_2$ models were compared to the SP and AP models. Wherever $\varphi > 0.95$ for one of the inference type families, the third analysis level was called upon. On this final level, surprise read-out functions were compared for the winning model class and corresponding inference type. The direct comparison of read-out models within the winning family allows for the use of protected exceedance probabilities (which are currently not available for family comparisons), which provide a robust alternative to inflated exceedance probabilities [66]. The step-wise procedure allows for spatio-temporal inference on particular read-out functions for which there is evidence for a belonging model class and inference type, facilitating the interpretation of the results. The hierarchical ordering thus moves from general to specific principles: the model class and inference type determine the probability estimates of the model, which are finally read out through surprise computation. While this procedure provides a plausible and interpretable approach to our model comparison, it should be noted that it constitutes an arbitrary choice in order to reduce data and model space and must be interpreted with caution. As a supplementary analysis, we performed non-hierarchical (factorial) family comparison analysis (S4 Fig) which groups the entire model space into the respective families for each family comparison without step-wise data reduction. The same procedure was used for the EEG sensor and source data.

To inspect the values of the forgetting-parameter $\tau$ that best fit the dipole data, subject specific free energy values were averaged across the timebins with surprise readout effects of interest for the corresponding dipoles. These were summed across subjects to yield the group log model evidence for each tested value of $\tau$, which were subsequently compared against each other.

**Model recovery study.** A simulation model recovery study was performed to investigate the ability to recover the models given the sequence data, model fitting procedure, and model comparison scheme. To this end, data was generated for n = 4000 (corresponding to the five concatenated experimental runs) by sampling from a GLM $y \sim N(X\beta, \sigma^2 I_n)$, after which model selection was performed. For the null-model, the design-matrix only comprised a column of ones. For all non-null models, an additional column of the z-normalized regressor was added. We set the true, but unknown $\beta_2$ parameter to 1, while varying $\sigma^2$, which function as the signal and noise of the data respectively. Given the z-scoring of the data, the $\beta_1$ parameter responsible for the offset is largely inconsequential and thus not further discussed. The model fitting procedure was identical to the procedure described in the supplementary material used for the EEG analyses (S2 Appendix).

For each noise level, we generated 40 data sets (corresponding to the number of subjects) to apply our random-effects analyses. This process was repeated 100 times for each of the different comparisons: null model vs DC model vs HMM (C1), DC $TP_1$ vs $TP_2$ (C2), DC SP vs AP vs $TP_1$ (C3), and DC $TP_1$ PS vs BS vs CS (C4). Family and model retrieval using exceedance probabilities worked well across all levels (S5 Fig), with a bias to the null model as signal-to-noise decreases. By inspecting the posterior expected values of $\beta_2$ and $\lambda^{-1}$ which resulted from fitting the model regressors to the EEG data, an estimate of the signal-to-noise ratio that is representative of the experimental work can be obtained. By applying the thresholds of $\varphi > 0.99$, $\varphi > 0.95$, $\varphi > 0.95$, and $\tilde{\varphi} > 0.95$ across the four comparisons respectively and subsequently inspecting the winning families and models at $\sigma^2 = 750$ (i.e., an SNR of 1/750), no false positives were observed. For C1 and C4, recovery was successful for all true, but unknown models in all of the 100 instances. While for C2 and to a lesser extent C3, concerning the families of

estimated sequence statistics, false negatives were observed only when confidence-corrected surprise was used to generate data. For C2, this led to false negatives in 67 ($TP_1$ CS) and 55 ($TP_2$ CS) percent of cases, while for C3 28 (SP CS), 0 (AP CS), and 33 ($TP_1$ CS) percent false negatives were observed. Each set of 40 data sets was generated with the same true, but unknown model. Due to the limited cognitive flexibility afforded by the distractor task, we did not expect large variability in the models used across subjects. Nevertheless, if this assumption is incorrect these simulations potentially overestimate the recoverability of the different models.

## Results

### Behavioural results and event-related potentials

Participants showed consistent performance in counting the amount of catch trials during each experimental run, indicating their ability to maintain their attention on the stimuli (robust linear regression of presented with reported targets: slope = 0.96, $p < 0.001$, $R^2 = 0.93$). Upon questioning during the debriefing, no subjects reported explicit awareness of switching regimes during the experiment.

An initial analysis was performed to confirm our paradigm elicited the typical somatosensory responses. Fig 6B shows the average SEP waveforms for contralateral (C4, C6, CP4, CP6) somatosensory electrodes with the expected evoked potentials, i.e. N20, P50, N140 and P300 resulting from stimulation of the left wrist. The corresponding topographic maps (Fig 6C) confirm the right lateralized voltage distribution of the somatosensory EEG components on the scalp. The EEG responses to stimulus mismatch were identified by subtracting the deviant from the standard trials (deviants-standards), thereby obtaining a difference wave for each electrode (see Fig 6D). The scalp topography of the peak differences between standards and deviants within predefined windows of interest indicates mismatch responses over somatosensory electrodes (Fig 6E).

To test for statistical differences in the EEG signatures of mismatch processing we contrasted standard and deviant trials with the general linear model. Three main clusters reached significance after performing family-wise error correction for multiple comparisons. The topographies of resulting F-values are depicted in Fig 7. The earliest significant difference between standard and deviant trials can be observed around 60ms post-stimulus (peak at 57ms, closest electrode CP4, $p_{FWE} = 0.002$, F = 27.21, Z = 5.07), followed by a stronger effect of the hypothesized N140 component around 120ms which will be referred to as the N140 mismatch response (N140 MMR, peak at 119ms, closest electrode: FC4, $p_{FWE} = 0.001$, F = 29.56, Z = 5.29). A third time window of a very strong and elongated difference effect starting around 250ms to 400ms post-stimulus which corresponds to the hypothesized P300 MMR (peak at 361ms, closest electrode: Cz, $p_{FWE} < 0.001$, F = 72.25).

The inspection of the $\beta$-parameter estimates at the reported GLM cluster peaks (illustrated in Fig 7) indicates that stimulus train length, i.e. the number of standard stimuli that precede a deviant stimulus, has differentiable effects on the size of EEG responses to standard and deviant stimuli. Both the N140 and P300 MMR effects are found to be parametrically modulated by train length as indicated by a significant linear relationship between $\beta$-estimates and train length. Specifically, the N140 MMR effect is reciprocally modulated by stimulus type, such that responses to standards are more positive for higher train lengths (F-statistic vs. constant model: 5.45, $p = 0.021$) while deviant responses become more negative (F-statistic vs. constant model: 5.07, $p = 0.026$). The parametric effect on the P300 MMR is entirely driven by the effect on deviant stimuli (F-statistic vs. constant model: 20.7, $p < 0.001$), with no effect of train

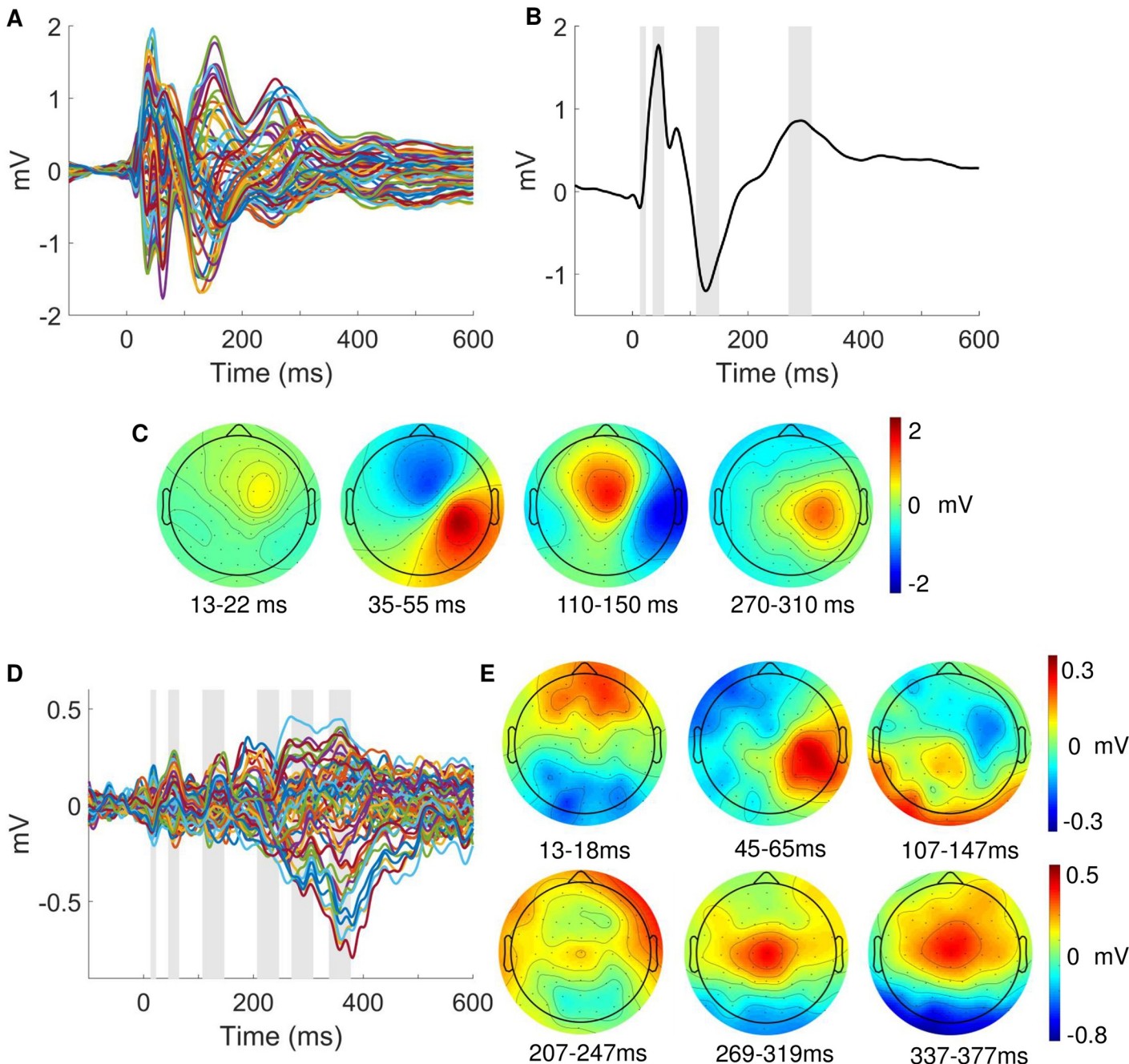

**Fig 6. Event-related potentials.** (A) Grand average SEP of all 64 electrodes. (B) Average SEP across electrodes C4, C6, CP4, CP6 (contralateral to stimulation). Grey bars indicate time windows around the standard somatosensory ERP components (13-23ms; 35-55ms; 110-150ms; 270-310ms). (C) ERP scalp topographies corresponding to the time windows in B. (D) Grand average ERP of the mismatch response obtained by subtraction of standard from deviant trials of 64 electrodes. Grey bars indicate windows around peaks which were identified within pre-specified time windows of interest around somatosensory ERP or expected mismatch response components (13-18ms; 45-65ms; 107-147ms; 207-247ms 269-319ms; 337-377ms). (E) ERP scalp topographies corresponding to the time windows in D.

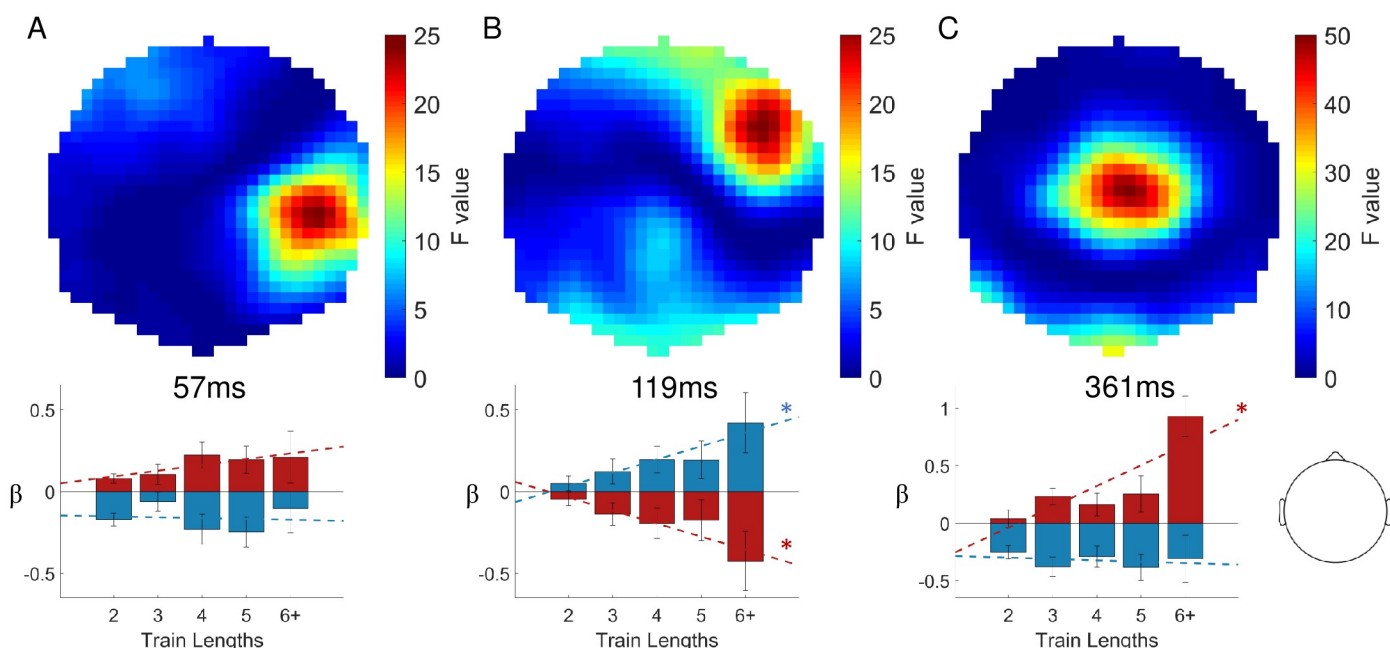

**Fig 7. Statistical parametric maps of mismatch responses.** Top row: Topographical F maps resulting from contrasting standard and deviant conditions averaged across the times of significant clusters: 57ms (A), 119ms (B) and 361ms (C). Bottom row: Corresponding beta parameter estimates of the significant peaks with deviants in red and standards in blue. Asterisks indicate significant linear fits ($p < 0.05$). Head depiction on the bottom right shows the orientation of the topographic maps.

length on the response to standard stimuli ($p > 0.05$). For the early 60ms cluster no effect was found on either standard or deviant stimuli.

## Source reconstruction

The distributed source reconstruction resulted in significant clusters at the locations of primary and secondary somatosensory cortex (Fig 8A, with details specified in the corresponding table). The resulting anatomical locations were subsequently used as priors to fit four equivalent current dipoles (Fig 8B, with details specified in corresponding table). Two dipoles were used to model S1 activity at time points around the N20 and the P50 components while an additional symmetric pair captured bilateral S2 activity around the N140 component. The moment posteriors of the S2 dipoles end up not strictly symmetric due to the soft symmetry constraints used by the SPM procedure [67].

To establish the plausibility of the somatosensory dipole model the EEG data was projected onto the four ECD's and the grand average source ERP was computed across subjects for standard and deviant trials. The resulting waveforms, shown in Fig 9, show a neurobiologically plausible spatiotemporal evolution: the two S1 dipoles reflect the early activity of the respective N20 and P50 components while the S2 dipoles become subsequently active and show strongest activity in right (i.e. contralateral) S2. The average response to standards and deviants within time windows around the significant MMR's in sensor space (around 57ms and 119ms; see Fig 7) were compared with simple paired t-tests. The S1$_{P50}$ dipole shows a significant difference at both time windows (at 57ms $p = 0.006$, $t = 2.94$; at 119ms $p = 0.009$, $t = 2.75$; bonferroni corrected) and can be suspected to be the origin of the effect at 57ms as well as contribute to

## A) Distributed Source Reconstruction

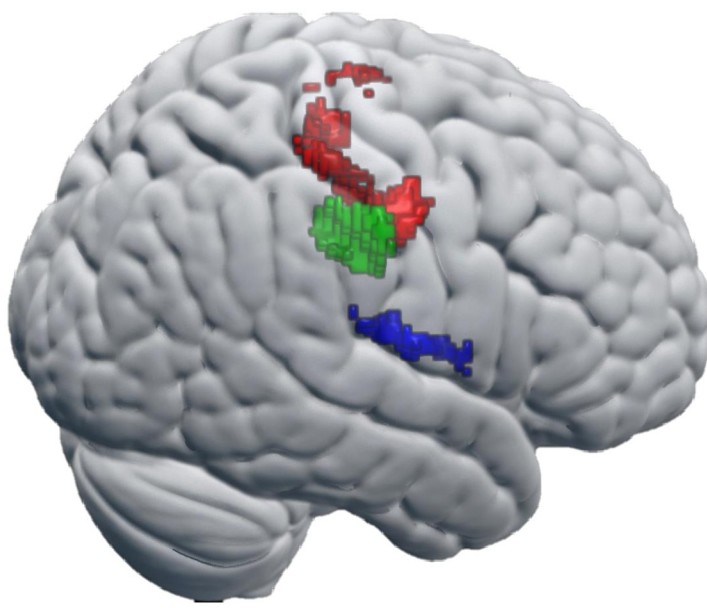

| Label | Time (ms) | p-Peak (FWE) | F value | MNI | Cytoarchitect. Reference |
|---|---|---|---|---|---|
| S1 (N20) | 18-25 | 0.019 | 30.23 | 44 -16 48 | Area 3b: 30% |
| S1 (P50) | 35-45 | <0.001 | 76.24 | 42 -20 38 | Area 3b: 37% Area 3a: 19% |
| S2 (right) | 110-160 | 0.019 | 29.85 | 62 -12 14 | Area OP4: 36% Area OP1: 23% |

## B) Equivalent Current Dipoles

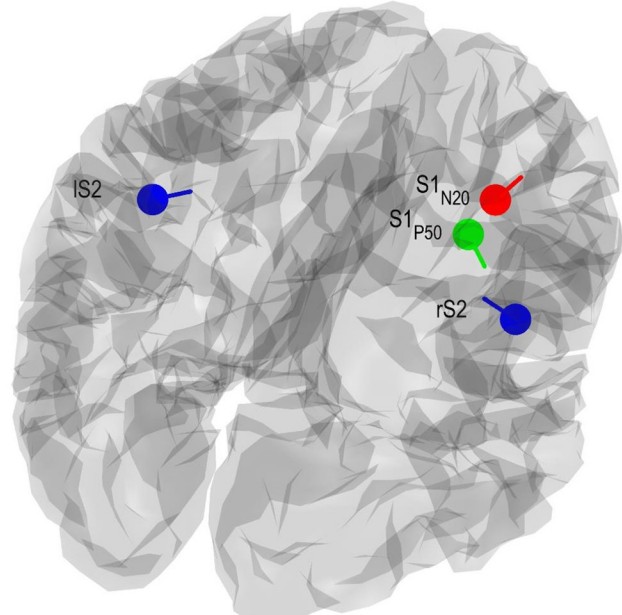

| Label | Time (ms) | Location Prior (MNI) | Location Posterior (MNI) | Moment Posterior |
|---|---|---|---|---|
| S1 (N20) | 20 | 44 -16 48 | 41 -12 53 | 2.92 1.67 3.89 |
| S1 (P50) | 45 | 42 -20 38 | 36 -17 44 | 17.31 -14.25 -6.49 |
| rS2 | 130 | 62 -12 14 | 46 -12 16 | -28.31 5.86 4.24 |
| lS2 | 130 | -62 -12 14 | -46 -12 16 | 12.69 5.86 4.24 |

**Fig 8. EEG source model.** (A) Statistical results of distributed source reconstruction. Red: 18-25ms, Green: 35-45ms, Blue: 110-160ms. Below: Table with corresponding detailed data of the clusters. (B) Location and orientation of fitted equivalent current dipoles. Red: S1 (N20), Green: S1 (P50), Blue: bilateral S2. Below: Table with their corresponding values.

the 119ms MMR while the right S2 dipole is mainly driving the strong 119ms effect ($p = 0.001$, $t = 3.44$; bonferroni corrected).

### Single trial modeling

We previously established the presence of mismatch responses in sensor space and confirmed their origin in the somatosensory system by modeling the early EEG components in source space. Subsequently, we investigated the temporal and spatial surprise signatures with trial-by-trial modeling of electrode and source data.

**Modeling in sensor space.** For large time windows at almost all electrodes there is strong evidence in favor of the DC model class ($\varphi > 0.99$), while the HMM model class does not exceed thresholding anywhere, therefore excluding HMM models from further analyses (Fig 10A). The corresponding threshold of expected posterior probabilities to arrive at comparable results lies around $\langle r \rangle > 0.75$ (see S6 Fig). To verify that this result was not merely due to an insufficient penalization of the DC models, the analysis was repeated with $\tau = 0$. Thus,

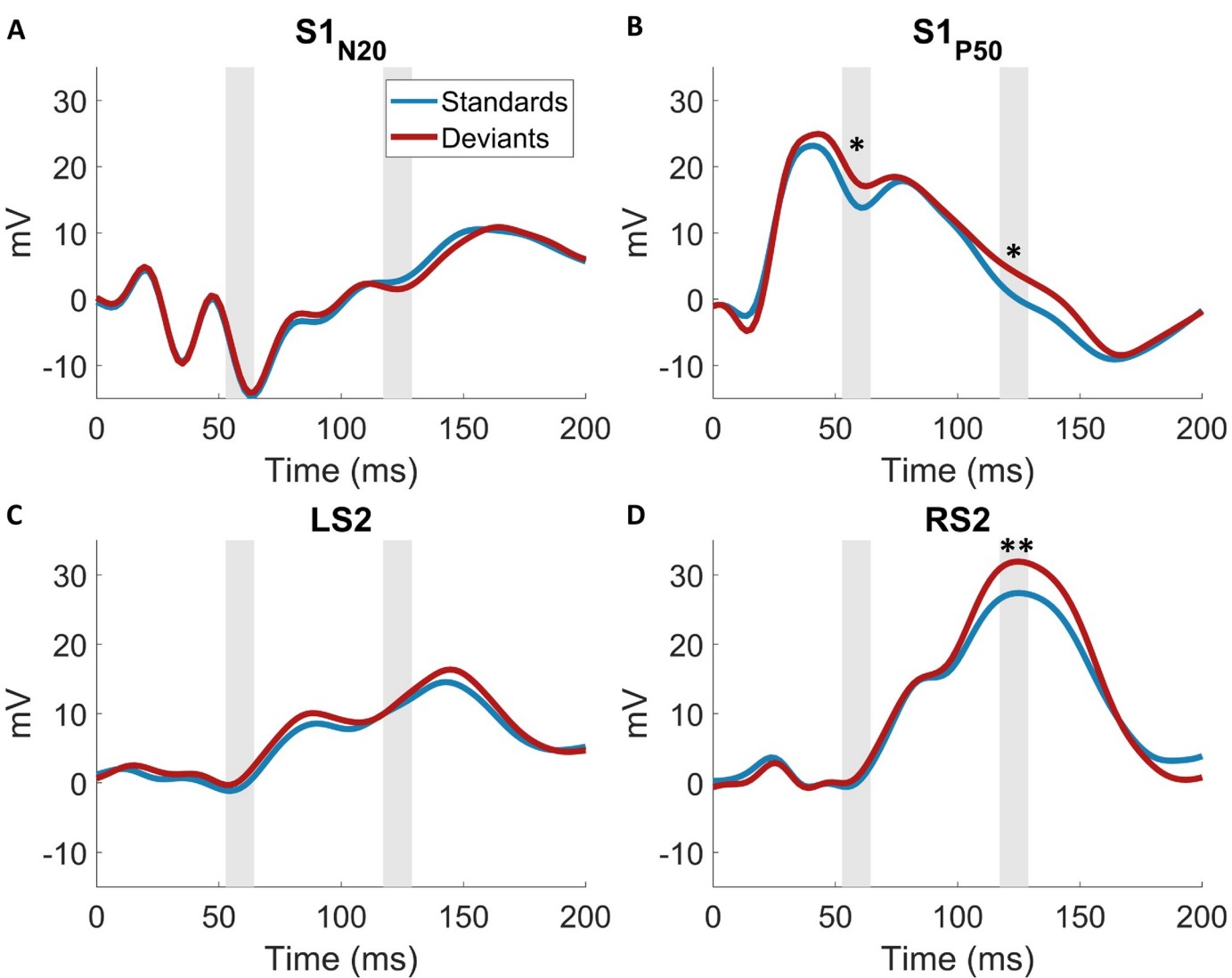

**Fig 9. Grand average waveforms of EEG dipole projections.** Standards and deviants were contrasted within time windows of interest informed by the GLM in the results section. $^*p < 0.05$; $^{**}p < 0.01$; Bonferroni corrected.

under this setting, all instances of the DC model had perfect, global integration similar to the HMM models. Likewise, no results above the threshold were found for the HMM model class (S7 Fig). Next, to ensure that the superiority of the DC model did not solely result from the additionally modeled catch trials, the HMM was compared with a DC model which did not capture these trials. This DC model still consistently outperformed the HMM, though it should be noted that the evidence for such a reduced DC model over the HMM is less pronounced (S6B Fig). For the DC model, $TP_1$ is found to outperform $TP_2$ ($\varphi > 0.95$, roughly corresponding to $\langle r \rangle > 0.7$), excluding $TP_2$ for the second and third level analyses. In the following step, $TP_1$ clearly performed better than SP and AP at almost all electrodes and time points (see Fig 10B and 10C; $\varphi > 0.95$, roughly corresponding to $\langle r \rangle > 0.7$). Thus, the following section presents the random-effects Bayesian model selection results of the readout functions of the Dirichlet-Categorical $TP_1$ model (shown in Fig 10D).

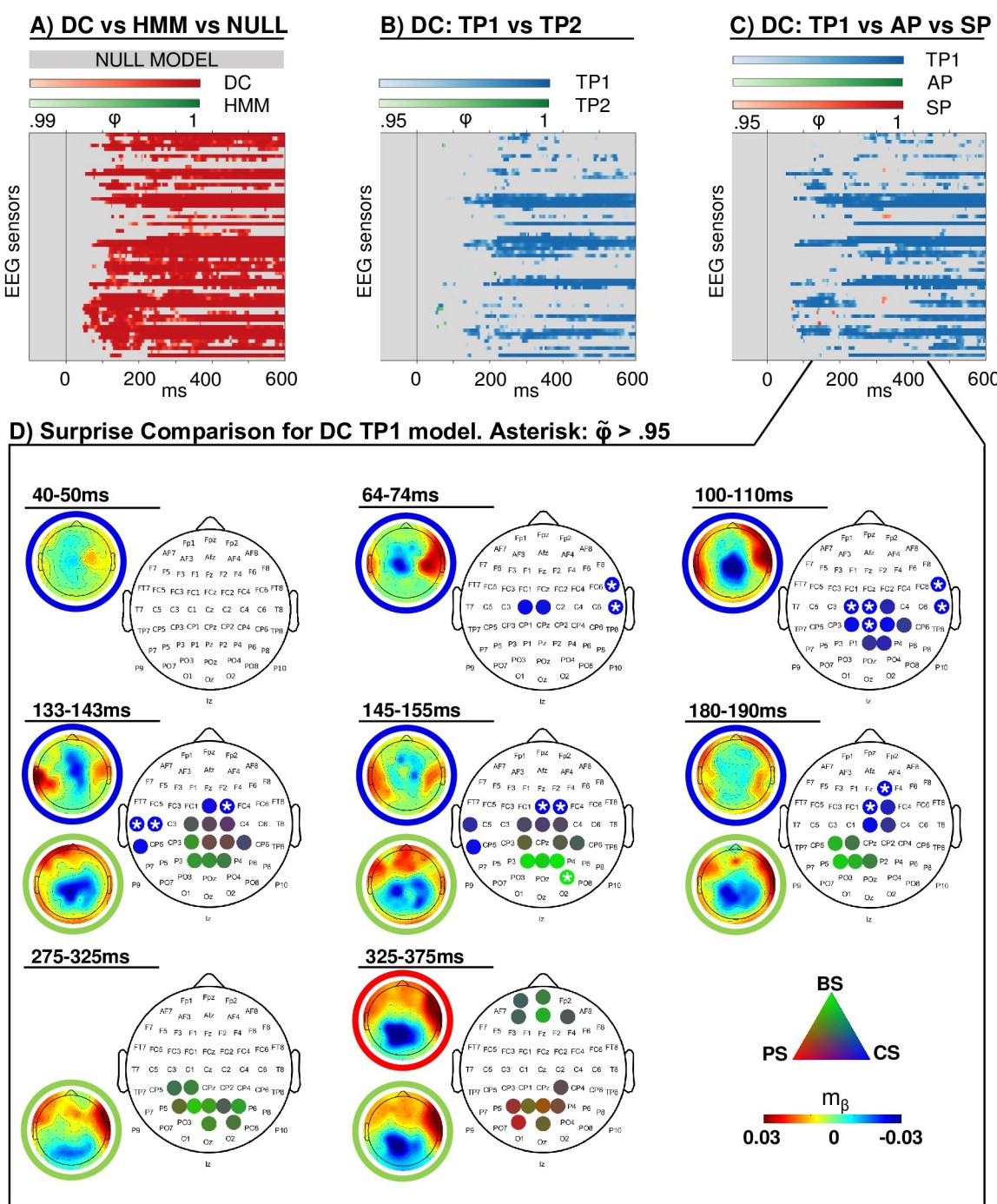

**Fig 10. Modeling results.** Exceedance probabilities ($\varphi$) resulting from the random-effects family-wise model comparison. (A) Dirichlet-Categorical (DC) model, Hidden Markov Model (HMM) and null model family comparison, thresholded at $\varphi > 0.99$ and applied for data reduction at all further levels. (B) Family comparison within the winning DC family, thresholded at $\varphi > 0.95$: first and second order transition probability models (TP$_1$, TP$_2$). (C) Family comparison within the winning DC family, thresholded at $\varphi > 0.95$: first order transition probability (TP$_1$), alternation probability (AP) and stimulus probability (SP) models and applied at the final level. (D) Unthresholded protected exceedance probabilities ($\tilde{\varphi}$) resulting from model comparison of surprise models within the winning DC TP1 family: Large discrete topographies show the electrode clusters of predictive surprise (PS) in red, Bayesian surprise (BS) in green and confidence-corrected surprise (CS) in blue. White asterisks indicate $\tilde{\varphi} > 0.95$ of single electrodes. Small continuous topographies display the converged variational expectation parameter $m_\beta$. This parameter may be interpreted as a $\beta$ weight in regression, indicating the strength and directionality of the weight on the model regressor that maximizes the regressor's fit to the EEG data (see S2 Appendix).

The scalp topographies depict the winning readout functions of the DC $TP_1$ model at different time windows. Given the difference in temporal dynamics of faster, early (<200 ms) and slower, late (e.g. P300) EEG components, different time windows were applied for averaging. Early clusters were identified by averaging protected exceedance probabilities over 10ms windows and using a minimum cluster size of two electrodes, while 50ms time windows were applied for averaging across later time windows with a minimum cluster size of four. The resulting clusters indicate that from around 70ms on, early surprise effects represented by confidence corrected surprise (CS) best explain the EEG data on contralateral and subsequently ipsilateral electrodes up to around 200ms. As demarcated in the plot, the early CS clusters include electrodes with $\tilde{\varphi} > 0.95$, which is indicative of a strong effect. A weaker cluster of Bayesian surprise (BS) is apparent at centro-posterior electrodes between 140-200ms of which the peak electrodes around 150ms show $\tilde{\varphi}$ between 0.8 and 0.95. As such, the mid-latency BS effect is less strong than the earlier CS clusters and can only provide indications. At the time windows of the P300 around 300 and 350ms, similar centro-posterior electrodes represent weak Bayesian surprise (peak $\tilde{\varphi}$ around 0.75) and predictive surprise (PS) clusters (peak $\tilde{\varphi}$ around 0.72), respectively. The mid-latency BS cluster is temporally in accordance with the putative N140 MMR while the late two clusters of BS and PS might be interpreted as indicative of a P300 MMR. However, especially the weak late clusters do not provide clear evidence in favour of a specific surprise readout function.

We note that the DC $TP_1$ vs $TP_2$ comparison in Fig 10B has few results prior to 200ms. This appears to fit with the model recovery study indicating that the least recoverable families are DC $TP_1$ and $TP_2$ in case of CS and the observation that CS is a winning surprise model for early time bins. In response, we conducted an additional family comparison between SP, AP, and TP encompassing both $TP_1$ and $TP_2$ (see S7 Fig). Clearly, more electrodes and time points with $\varphi > 0.95$ can be observed in the early time window, suggesting that early effects are driven by TP inference but that for empirical data, we are unable to convincingly resolve $TP_1$ and $TP_2$ for CS computation. Furthermore, it should be noted that our step-wise model comparison approach constitutes a reasonable, yet arbitrary choice to create summary statistics of our data set and a large model space. In an additional analysis, we performed a non-hierarchical model comparison which grouped the entire model space in the respective families of interest without step-wise data reduction. These results (S4 Fig) broadly replicate the findings from the hierarchical approach across the levels and likewise indicate that the order of transition probability (TP1 and TP2) can not be resolved in early time windows.

**Modeling in source space.** The topographic distribution of the effects of confidence-corrected surprise seem to indicate an early contribution of secondary somatosensory cortex from around 70ms on that starts contra- and extends ipsilaterally while the weaker BS cluster emerges around the time of the N140 MMR. In order to further investigate this observation and examine the spatial origins of the surprise clusters, we fit our models to the single trial dipole data and used the same hierarchical Bayesian model selection approach as for the sensor-space analysis described in the Materials and Methods section. Results for the source activity were highly similar, with clear results in favour of the DC and $TP_1$ model families at thresholds of $\varphi > 0.99$ and $\varphi > 0.95$, respectively. Consequently, the surprise readout functions of the DC $TP_1$ model were subjected to BMS. The results depicted in Fig 11 support the interpretation of an early onset of CS in secondary somatosensory cortex ($\tilde{\varphi} > 0.95$) and allocate the later onset BS cluster in electrode space to primary somatosensory cortex ($\tilde{\varphi}$ ranging from around 0.7 to 0.9). However, as is also apparent in electrode space, this mid-latency BS effect is weak and can only provide an indication.

**Leaky integration.** We inspected the $\tau$-parameter values that resulted in the highest group log model evidence for the reported dipole effects (Fig 11). All three considered clusters

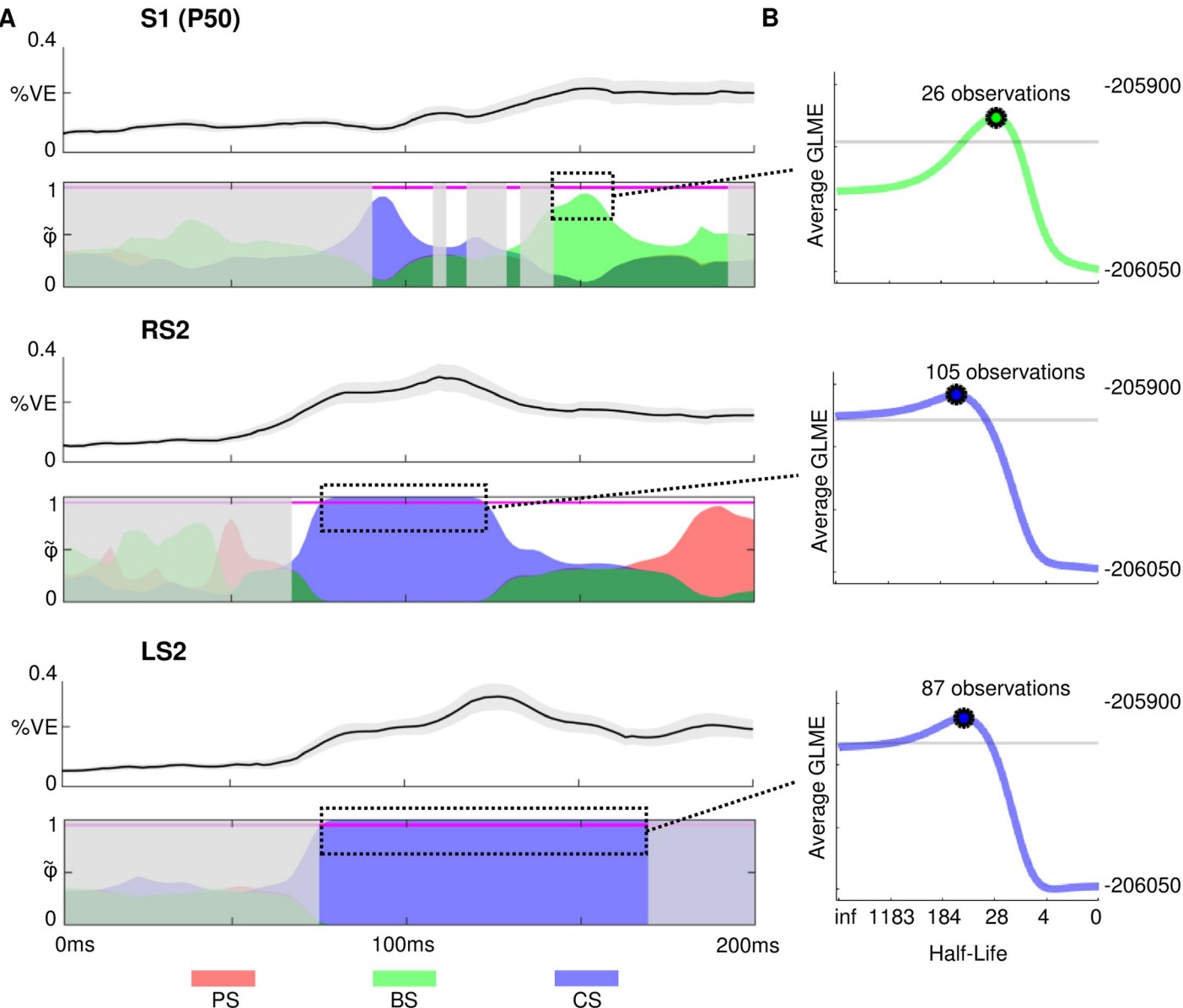

**Fig 11. Modeling results in source space with best fitting forgetting-parameter values.** Red: Predictive surprise (PS), Green: Bayesian surprise (BS), Blue: Confidence-corrected surprise (CS) A) Colored areas depict protected exceedance probabilities ($\tilde{\varphi}$) of the surprise readout functions of the Dirichlet-Categorical TP$_1$ model within the dipoles S1P50, right S2 (RS2) and left S2 (LS2) using alpha blending. In grey shaded areas the DC model family shows $\varphi < 0.99$ or the TP$_1$ model family $\varphi < 0.95$. The S1N20 dipole was omitted in the visualization as no model is observed above this threshold. Magenta horizontal lines indicate $\tilde{\varphi} = 0.95$. Line plots above each dipole plot show the respective mean percent variance explained of the models in dotted rectangles ± standard error. B) The group log model evidence (GLME) values corresponding to the stimulus half-lives for forgetting-parameter $\tau$, after averaging the timebins inside the dotted-rectangles (S1P50: 145-191ms; RS2: 68-143ms; LS2: 76-168ms). The grey lines indicate a difference of 20 GLME from the peak, indicating very strong evidence in favour of the peak half-life value compared to values below this threshold.

indicate a local timescale of integration, with the best-fitting parameter values resulting in a stimulus half-life of $\sim 105$ and $\sim 87$ for the confidence-corrected surprise effects at 75-120ms and 75-166ms respectively, and a half-life of $\sim 26$ observations for the Bayesian surprise effect at 143-157ms. Using the single-subject peaks, $\tau$ was found to significantly differ from 0 (i.e., no

forgetting) for the BS effect in S1 ($p < 0.001$) and CS in RS2 ($p < 0.05$), but not in LS2 (p = 0.06). Paired t-tests revealed no significant difference in $\tau$ underlying the three effects ($p > 0.05$).

## Discussion

In this study, we used a roving paradigm to identify EEG mismatch responses independent of stimulus identity. The early MMR effects were source localized to the somatosensory system and the N140 and P300 MMR's show differential linear dependence on stimulus train lengths for standard and deviant stimuli. Using computational modelling, EEG signals were best described using a non-hierarchical Bayesian learner performing transition probability inference. Furthermore, we provide evidence for an early representation of confidence-corrected surprise localized to bilateral S2 and weak indications for subsequent Bayesian surprise encoding in S1. These computations were shown to use a local, rather than global, timescale of integration.

We report a significant somatosensory mismatch response around three distinct post-stimulus time-points: 57ms, 119ms, and 361ms. These will be referred to as sMMR's as opposed to MMN since the effects at 57ms and 361ms are not negativities and our experimental protocol included an explicit attentional focus on the stimulation. The MMN was originally defined to be a pre-attentive effect and while attention to the stimulus does not seem to influence the MMN in the visual domain [68], we don't address a potential independence of attention here. Nevertheless, the reported sMMR effects integrate well with previous findings on the somatosensory MMN (sMMN). Our 119ms effect is in line with the timing of the most commonly reported sMMN as a modulation of the N140 component between 100-250ms [17, 21, 22, 23, 24, 25, 26, 27, 28, 29, 30, 31]. However, some studies additionally describe a modulation of multiple somatosensory components [17 18, 19, 24], similar to our three distinct sMMR effects. The electrode positions reported in sMMN studies show a large variability of frontocentral and parietal electrodes. These discrepancies might be driven by the differences in stimulation sites (different fingers and hand) and deviant definitions (vibrotactile frequencies, stimulation site, stimulation duration). Here, we present significant effects around C4 and FC4 electrodes for the 57 and 119ms time-points, respectively, indicating EEG generators within the contralateral somatosensory system. This implication is in line with intracranial EEG recordings of the somatosensory cortex during oddball tasks [24, 30]. In accordance with previous MEG studies using source localization [21, 22], our source space analysis suggests the early MMR effects to originate from contralateral primary and secondary somatosensory cortex (cS1 and cS2, respectively), with the earliest MMR (at 57ms) localized to cS1 followed by a combined response of S1 and S2. While evidence exists for a role of S2 in the early phase of mismatch processing [26], the evolution from an initial MMR generated by S1 to an additional involvement of S2 in the mid-latency MMR, as indicated by our findings, is consistent with the sequential activation of the somatosensory hierarchy in general tactile stimulus processing [69, 70, 71]. Finally, the third sMMR effect at 361ms is in accordance with a large body of evidence showing a modulation of the P300 component by mismatch processing [72, 73, 74]. The P300 in response to oddball tasks likely reflects a modality unspecific effect, dependent on task-related resource allocation [75, 76, 77, 78, 79] and contingent on attentional engagement [29].

In addition to three spatiotemporally distinct sMMR effects, we further show their differential modulation by the length of standard stimulus trains preceding the deviant stimulus. This finding supports the interpretation that distinct mechanisms underlie the generation of the different sMMR's. The earliest effect around 57ms is not affected by train length, possibly

reflecting a basic change detection mechanism that signals a deviation from previous input regardless of statistical regularities. The mid-latency MMR around 119ms, on the other hand, shows a significant linear dependence on stimulus train length for both deviant and standard stimuli. Longer train lengths result in parametrically stronger negative responses to deviant stimuli while responses to standard stimuli are increasingly reduced. This effect is in accordance with repetition suppression effects reported for the MMN [80, 81] which have been shown to be dependent on sequence statistics and are interpreted to reflect perceptual learning [82, 83]. While it has been indicated that the number of preceding standards can also enhance the sMMN [26], no previous studies show comparable effects to our parametric modulation of the mid-latency sMMR. The reciprocal effect of repetition for standard and deviant stimuli shown here indicate early perceptual learning mechanics in the somatosensory system, likely originating from S2 in interaction with S1. In contrast, later mismatch processing reflected by the sMMR at 361ms only shows a linear dependence of deviant stimuli on train length, while the response to standards remains constant. This is in line with the interpretation that perceptual learning in the P300 reflects a recruitment of attention in response to environmental changes, possibly accompanied by updates to this attentional-control system [41].

In addition to average-based ERP analyses, single-trial brain potentials in response to sequential input can provide a unique window into the mechanisms underlying probabilistic inference in the brain. Here, we investigated the learning of statistical regularities using different Bayesian learner models with single-trial surprise regressors. Partitioning the model space allowed us to infer on distinguishing features between the model families using Bayesian model selection (BMS). The first comparison concerned the form of hidden state representation: In order for a learner to adequately adapt one's beliefs in the face of changes to environmental statistics, more recent observations may be favored over past ones without modeling hidden state dynamics (Dirichlet-Categorical model; DC), or different sets of statistics may be estimated for a discretized latent state (Hidden Markov Model; HMM). Our comparison of these two learning approaches provides strong evidence for the DC model class over the HMM for the large majority of electrodes and post-stimulus time. The superiority of the DC model was found to be irrespective of the inclusion of leaky integration to the DC model, indicating the advantage of a non-hierarchical model in explaining the EEG data. It is noteworthy that part of the strength of the DC model depended on the modelling of the catch trial, although a reduced DC model still outperformed the HMM. Participants were neither aware of the existence of the hidden states in the data generation process, nor was their dissociation or any tracking of sequence statistics required to perform the behavioural task. As such, the early EEG signals studied here are likely to reflect a form of non-conscious, implicit learning of environmental statistics [84, 85, 86]. However, it is possible that the brain implements different learning algorithms in different environments, resorting to more complex ones only when the situation demands it. As the discrete hidden states produced relatively similar observation sequences, more noticeable transitions between hidden states may provide an environment with greater incentive to implement a more complex model to track these states, which might have yielded different results. Indeed, humans seem to assume different generative models in different contexts, possibly depending on task instructions [87]. This may in part explain why evidence has been provided for the use of both hierarchical [88, 89] and non-hierarchical models [90, 91]. Nevertheless, it has been suggested that the brain displays a sensitivity to latent causes in low-level learning contexts [92], which might indicate the relevance of other factors. For example, it is possible the currently tested HMM may be too constrained and a simpler, more general changepoint-detection model [89] may have performed better. By omitting instructions to learn the task-irrelevant statistics, our study potentially avoids the issue of invoking a certain generative model. We might therefore report on a 'default' model of the

brain used to non-consciously infer environmental statistics. The sort of computations (relating to surprise and belief updating) and learning models we consider might be viewed in light of theories such as predictive coding and the free energy principle for which preliminary work suggests implementational plausibility (e.g. [93]). The computation models tested in the current study do not provide a biophysically plausible manner by which the brain acquires the estimated transition probabilities and subsequent surprise quantities. Rather, the models serve to identify qualities that a future successful biophysically plausible algorithm should exhibit.

In order to investigate which statistics are estimated by the brain during the learning of categorical sequential inputs, we compared three models within the DC model family that use different sequence properties to perform inference on future observations: stimulus probability (SP), alternation probability (AP), and transition probability (TP) inference. The TP model subsumes SP and AP models and is thus more general by maintaining a larger hypothesis space. Our results show that the TP model family clearly outperformed the SP and AP families, suggesting that the brain captures sequence dependencies by tracking transitions between types of observations for future inference. We thereby provide further evidence for an implementation of a minimal transition probability model in the brain as recently concluded from the analysis of several perceptual learning studies [94], extending it to include somesthesis. Additionally, we expand upon previous studies by comparing a first order TP model ($TP_1$), capturing transitions between stimuli conditional only on the previous observation, with a second order TP model ($TP_2$), which tracks transitions conditional on the past two observations. Our results suggest that the additional complexity of the second order dependencies contained in our stimulus sequence were not captured by the brain, although we were not able to convincingly show this for early CS computation. Nevertheless, the brain may resort to alternative, more compressed representations [95].

The BMS analyses of the partitioned model space suggests that the brain's processing of the stimulus sequences is best described by a Bayesian learner with a static hidden state (akin to the DC model) which estimates first-order transition probabilities ($TP_1$). Within the DC-$TP_1$ model family, we compared the surprise quantifications themselves as the readout functions for the estimated statistics of the Bayesian learner: predictive surprise (PS), Bayesian surprise (BS), and confidence-corrected surprise (CS). The results indicate that the first surprise effect is represented by CS from around 70ms over contralateral somatosensory electrodes which extends bilaterally and dissipates around 200ms. BS is found as a second, weaker centro-posterior electrode cluster of surprise between 140-180ms. As proposed by Faraji et al. [35], CS is a fast-computable measure of surprise in the sense that it may be computed before model updating occurs. In contrast, as BS describes the degree of belief updating, which requires the posterior belief distribution, it is expected to be represented only during the update step or later. As such, the temporal evolution of the observed CS and BS effects is in accordance with the computational implications of these surprise measures. Specifically, our study provides support for the hypothesis that the representation of CS, as a measure of puzzlement surprise, precedes model updating and may serve to control update rates. While PS is also a fast-computable puzzlement surprise measure and (similarly to CS) is scaled by the subjective probability of an observation, CS additionally depends on the confidence of the learner, read out as the (negative) entropy of the model. Evidence for a sensitivity to confidence of prior knowledge in humans has been reported in a variety of tasks and modalities [96, 97, 98]. This further speaks to the possibility that CS informs belief updating, as confidence has been suggested to modulate belief updating for other modalities in the literature [99, 100] and is explicitly captured in terms of belief precision by other promising Bayesian models [101, 102, 103]. We suspect that, similarly, confidence concerns the influence of new observations on current beliefs in somatosensation. However, as this was not explicitly modelled and investigated in

the current work we were not able to test it directly. Furthermore, as the state transition probability between regimes was fixed in the current study, it is not well suited to address the effects of the volatility of the environment on belief updating. Future work might focus on the interplay of environmental volatility and confidence in their effects on the integration of novel observations. It is important to note that one may also be confident about novel sensory evidence (e.g. due to low noise) which may result in larger model updates [104]. This aspect of confidence, however, lies outside the scope of the current work.

Our source reconstruction analyses attributed the early CS effects to the bilateral S2 dipoles, which is in accordance with the timing of S2 activation reported in the literature [69, 70, 71]. This finding suggests that the secondary somatosensory cortex may be involved in representing confidence about the internal model. The BS effect around 140ms was less pronounced in source space only peaking at $\tilde{\varphi}$ of 0.89 and was localized to S1. Despite the weak evidence for this BS representation around a 140ms somatosensory MMR, its timing matches prior work using modeling of Bayesian surprise signals in the somatosensory system [13]. Generally, our findings are in accordance with previous accounts of perceptual learning in the somatosensory system [105]. In sum, these results suggest that the secondary somatosensory cortex may represent confidence about the internal model and compute early surprise responses, potentially controlling the rate of model updating. Signatures of such belief updating, were found around the time of the N140 somatosensory response and were localized to S1. Together, these effects might be interpreted as a possible interaction between S1 and S2 that could be responsible for both a signaling of the inadequacy of the current beliefs and their subsequent updating.

In an attempt to relate the surprise readouts to the mismatch responses, we averaged surprise regressors to obtain model-based predictions for the standard-deviant contrasts. First, all $TP_1$ models except HMM CS predict the existence of an MMR, i.e., a difference in the averaged response between standard and deviant trials. Second, for multiple models an increase in train length leads to reduced surprise to standards and increased surprise to deviants. The CS readout is scaled by PS and BS, as well as by belief commitment, which increases for standards and decreases for deviants. This counteracting effect of belief commitment and the surprise terms can lead to independence of CS and train length when responses are averaged, manifesting in the current sequences only for standard trials. As the early MMR was found to be independent of train length, this indicates a possibility for a potential relation between these results. The intermediate MMR roughly temporally co-occurs with a simultaneous representation of BS and CS in S1 and S2. The dependence of the mid-latency MMR on train-length for both standards and deviants and the encoding of belief inadequacy and updating quantities is suggestive of convergent support in favor of a perceptual learning response which involves both somatosensory cortices. DC BS is however not the only model which predicts this dependence, highlighting the reduced ability to distinguish between models by averaging trials. At the P300 MMR it was found that only the response to deviants is dependent on train length. The averaged response of DC CS is most compatible with this ERP, however, this is unlikely to be meaningful as the model was not found to fit the single-trial EEG data well around this time. It is noteworthy that belief updating as described by DC BS, which is best describing the EEG data around that time, does not accurately predict the ERP dynamics of the P300, which matches the relative weakness of the BS effect in the single-trial EEG analysis. While a role of the P300 response in Bayesian updating has previously been reported [13, 40], the currently presented P300 dynamics may better be captured by alternate accounts, such as a reflection of an updating process of the attention allocating mechanism as suggested by Kopp and Lange [106].

Our implementation of the Dirichlet-Categorical model incorporates a memory-decay parameter $\tau$ that exponentially discounts observations in the past. The $\tau$-values for the winning models of our BMS analyses that best fit the data for the surprise effects of interest

indicate relatively short integration windows for both CS and BS with stimulus half-lives of approximately 95 and 26 observations, respectively. This suggests that, within our experimental setup, the brain uses local sequence information to infer upcoming observations rather than global integration, for which all previous observations are considered. For a sub-optimal inference model with a static hidden state representation, the incorporation of leaky observation integration on a more local scale can serve as an approximation to optimal inference resorting to dynamic latent state representation and can thereby capture a belief about a changing environment [94].

Given a very large timescale, BS converges to zero as the divergence between prior and posterior distributions decreases over time, imposing an upper bound on the timescale. Meanwhile, for PS and CS it tends to lead to more accurate estimates of $p(y_t|s_t)$ as more observations are considered. However, given the regime switches in our data generation process, a trade-off exists where a timescale that is too large prevents flexible adaptation following such a switch. In the current context, the timescales are local enough where the estimated statistics are able to be adapted in response to regime switches (with a switch occurring every 100 stimuli on average). Especially CS shows a large range of $\tau$-values producing similarly high model evidence due to the high correlation between regressors. In sum, it is possible that the same timescale is used for the computation of both the CS and BS signals, as the differences in optimal $\tau$-values between clusters were not found to be significant. This interpretation is most intuitively compatible with the hypothesis that the early surprise signals may control later belief updating signals. Although the uncertainty regarding the exact half-lives is in line with the large variability found in the literature, local over global integration is consistently reported [9, 13, 39, 48, 94, 95]. Given a fixed inter-stimulus interval of 750ms, a horizon of 95 and 26 observations may be equated to a half-life timescale of approximately 71 to 20 seconds, with regime switches expected to occur every 75 seconds.

Some considerations of the current study deserve mention. First, the behavioural task required participants to make a decision about the identity of the stimulus so as to identify target (catch) trials. Thus, one may wonder to what extent the results contain conscious decision making signals, rather than implicit, non-conscious learning activity. However, decision making-related signals are described to occur relatively late in the trial [107, 108] and we assume to largely avoid them here by focusing on early signals prior to 200ms. Secondly, a large model space of both hierarchical as well as non-hierarchical Bayesian learners exists. As such, it is possible that the brain resorts to some hierarchical representation different from the ones tested here. We chose to use an HMM as it closely resembled the underlying data structure, offers the optimal solution for a discrete state environment, and contributes to the field as it has seen only limited application for probabilistic perceptual learning. Furthermore, some limitations might concern the stepwise model comparison intended to yield interpretable results by allowing inference on the generative model giving rise to surprise signatures. A reduction of both data and model space is not a standard procedure in Bayesian model comparison and we stress that we do not provide a methodological validation of this approach. Nevertheless, we argue that this scheme capitalizes on the hierarchical structure of the model space, provide model recoverability simulations, and present similar results using a standard factorial family comparison to support that the main conclusions are not dependent on the exact model comparison approach. The analyses performed here include a large number of independent Bayesian model comparisons (as is not uncommon in neuroimaging), yet no corrections are applied. While the resulting exceedance probabilities are reported here only above a given threshold, these model comparisons do not constitute statistical tests per se, as they do not provide a mapping from the data to binary outcomes. It follows that the analyses do not suffer from a classical multiple testing problem, which can be addressed using the control of multiple

testing error rates (e.g. the control of the family-wise error rate for fMRI inference based on random field theory). Nevertheless, it would be valuable for methodological advances to consider the possibility of randomly occurring high exceedance probabilities given a large number of independent model comparisons. A multilevel scheme which adjusts priors over models, rather than the current ubiquitous use of flat priors, may be developed as a satisfactory approach [109, 110, 111]. As the current method is agnostic to the large number of model comparisons we need to stress that we only report preliminary evidence.

In conclusion, we show that signals of early somatosensory processing can be accounted for by (surprise) signatures of Bayesian perceptual learning. The system appears to capture a changing environment using a static latent state model that integrates evidence on a local, rather than global, timescale and estimates transition probabilities of observations using first order dependencies. In turn, we provide evidence that the estimated statistics are used to compute a variety of surprise signatures in response to new observations, including both puzzlement surprise scaled by confidence (CS) in secondary somatosensory cortex and weak indications for enlightenment surprise (i.e. model updating; BS) in primary somatosensory cortex.

## Supporting information

**S1 Appendix. Bayesian learner models.** In this supplementary text we provide the derivations for the presented equations of the compared Bayesian learner models.
(PDF)

**S2 Appendix. A free-form variational inference algorithm for general linear models with spherical error covariance matrix.** In this supplementary text we present the algorithm used to approximate log model evidence for subsequent Bayesian model comparison.
(PDF)

**S1 Fig. Estimated emission probabilities and latent regime inference of the hidden Markov model.** (A) The average emission probabilities of the stimulus probability (SP), alternation probability (AP), and transition probability (TP) hidden Markov model (HMM) for both states ($s$) at the final timestep of each sequence. For $TP_2$, a comparison is provided of the emission probabilities used for data generation and the average, normalized emission probabilities estimated by the HMM. Error bars represent the standard error of the mean. (B) Correlating the true regimes and filtering posterior over time confirms that AP and TP inference allow for the tracking of the fast and slow-switching regimes, while SP inference does not capture the necessary dependencies due to the regimes being balanced in terms of stimulus probabilities.
(TIF)

**S2 Fig. Model-derived predictions for standard and deviant stimuli.** Averaged surprise readouts using either the (left) 25000 total sequences or (right) 200 sequences administered to the participants elicited for standard and deviant stimuli following a certain amount of repeating stimuli (train length). The model-derived predictions are relatively well-preserved in the smaller data-set. Only first-order transition probability models are plotted. Error bars indicate standard deviations. The used stimulus half-lives of 95 and 26 are representative of the winning models in the single-trial EEG analysis. DC: Dirichlet-Categorical model; HMM: Hidden Markov Model; PS: Predictive surprise; BS: Bayesian surprise; CS: Confidence-corrected surprise; No F: model without forgetting (i.e. perfect integration); HL: stimulus half-life.
(TIF)

**S3 Fig. Schematic of the hierarchical approach to family-wise Bayesian model selection.** First level (depicted in the top row): The 12 DC models and the 12 HMM models were

grouped into their corresponding model class family and compared via BMS against each other and an offset Null-Model. Second level (lower row, left rectangle): Within the DC model class, the two transition probability models $TP_1$ and $TP_2$ were grouped into families and the winner of the BMS was used for the comparison against the other two inference type models (Stimulus Probability (SP) and Alternation Probability (AP)). Third Level (lower row, middle rectangle): The surprise readouts of the DC $TP_1$ model were subjected to BMS and the resulting exceedance probabilities are reported in the main results. Thresholding of the model class families and inference types was again applied at successive levels leading to data reduction. (TIF)

**S4 Fig. Non-hierarchical family-wise Bayesian model selection.** Exceedance probabilities ($\varphi$) resulting from the RFX family model comparison by investigating the full model space in each comparison. A) Family comparison of the first order transition probability ($TP_1$), second order transition probability ($TP_2$), alternation probability (AP; no above-threshold results with $\varphi > 0.95$) and stimulus probability (SP) models; thresholded at $\varphi > 0.95$. B) Unthresholded family comparison of surprise models. Large discrete topographies show the electrode clusters of predictive surprise (PS) in red, Bayesian surprise (BS) in green and confidence-corrected surprise (CS) in blue. White asterisks indicate $\varphi > 0.95$. Small continuous topographies display the converged variational expectation parameter ($m_\beta$). (TIF)

**S5 Fig. Model recovery study.** A model recovery study was performed using simulated data. Subplots (A-D) show the average exceedance probabilities (shading represents standard deviations) of 100 random-effects Bayesian model selection analyses under different signal-to-noise ratios. This was performed for (A) Null Model vs DC Model vs HMM families, (B) DC $TP_1$ vs $TP_2$ families, (C) DC SP vs AP vs $TP_1$ families, and (D) DC $TP_1$ PS, BS, and CS models. Noteworthy is that the instances of reduced differentiability for (B) and (C) occurred only when the true, but unknown model was confidence-corrected surprise. (E) An estimate of the signal-to-noise of the experimental single-trial EEG analyses by inspecting the ratio of the expected posterior estimates of the model fitting procedure for $\beta^2$ and $\lambda^{-1}$. (TIF)

**S6 Fig. Expected posterior probabilities of hierarchical Bayesian model-selection.** Expected posterior probabilities ($\langle r \rangle$) resulting from family model comparisons. A) Dirichlet-Categorical (DC) model, Hidden Markov Model (HMM) and Null model family comparison, thresholded at $\langle r \rangle > 0.75$. B) Family comparison within the winning DC family, thresholded at $\langle r \rangle > 0.7$: first and second order transition probability models ($TP_1$, $TP_2$). C) Family comparison within the winning DC family, thresholded at $\langle r \rangle > 0.7$: first order transition probability ($TP_1$), alternation probability (AP) and stimulus probability (SP) models. (TIF)

**S7 Fig. Additional random effects family-wise comparisons.** (A) Comparison of the model families: Null model, Dirichlet-Categorical model (DC) with tau = 0 (i.e. no forgetting and no penalization) and Hidden Markov Model (HMM). (B) Comparison of the model families: Null model, DC without modelling the catch trials and HMM. (C) Comparison of the model families: Null model, DC with and DC without modelling the catch trials. (D) Comparison of the model families within the DC model: Stimulus probability model (SP), alternation probability model (AP) and transition probability model family (TP) subsuming first and second order TP models in one family. Exceedance probabilities ($\varphi$) are plotted for all comparisons. (TIF)

## Acknowledgments

The authors would like to thank the HPC Service of ZEDAT, Freie Universität Berlin, for computing time.

## Author Contributions

**Conceptualization:** Sam Gijsen, Miro Grundei, Robert T. Lange, Dirk Ostwald, Felix Blankenburg.

**Data curation:** Sam Gijsen, Miro Grundei.

**Formal analysis:** Sam Gijsen, Miro Grundei.

**Funding acquisition:** Felix Blankenburg.

**Investigation:** Sam Gijsen, Miro Grundei.

**Methodology:** Sam Gijsen, Miro Grundei, Robert T. Lange, Dirk Ostwald.

**Project administration:** Sam Gijsen, Miro Grundei, Felix Blankenburg.

**Resources:** Felix Blankenburg.

**Software:** Sam Gijsen, Miro Grundei, Robert T. Lange, Dirk Ostwald.

**Supervision:** Dirk Ostwald, Felix Blankenburg.

**Validation:** Sam Gijsen, Miro Grundei.

**Visualization:** Sam Gijsen, Miro Grundei, Robert T. Lange.

**Writing – original draft:** Sam Gijsen, Miro Grundei, Robert T. Lange.

**Writing – review & editing:** Sam Gijsen, Miro Grundei, Dirk Ostwald, Felix Blankenburg.

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
