## [Decision Letter · Decision Letter 0]

21 Jul 2020

Dear Dr Gijsen,

Thank you very much for submitting your manuscript "Neural surprise in somatosensory Bayesian learning" for consideration at PLOS Computational Biology.

As with all papers reviewed by the journal, your manuscript was reviewed by members of the editorial board and by several independent reviewers. In light of the reviews (below this email), we would like to invite the resubmission of a significantly-revised version that takes into account the reviewers' comments.

Your manuscript has been favorably reviewed by three reviewers and they all agreed that this study makes a valuable contribution but also suggested some major revisions alongside several minor issues for clarification. In particular, the reviewers raised several concerns regarding the nature of the model comparison and asked for several clarifications regarding the analyses and interpretation of the results. It would be important to address those issues in a revised manuscript.

We cannot make any decision about publication until we have seen the revised manuscript and your response to the reviewers' comments. Your revised manuscript is also likely to be sent to reviewers for further evaluation.

Sincerely,

Philipp Schwartenbeck

Guest Editor

PLOS Computational Biology

Samuel Gershman

Deputy Editor

PLOS Computational Biology

Reviewer's Responses to Questions

**Comments to the Authors:**

Reviewer #1: Thank you for asking me to review this ms. It represents a methodologically robust approach to a question of great relevance for systems and computational neuroscience (neural and computational mechanisms underlying statistical learning). My comments are mainly related to the interpretation of the findings.

Study outline

- Gijsen and colleagues investigate use a somatosensory roving-stimulus paradigm in conjunction with EEG to investigate how the somatosensory system tracks the statistical regularities of sensory stimuli. The paradigm employed sequences of low and high intensity tactile stimuli of variable length, and two hidden states (governing whether the switching between stimulus types was ‘’fast’ or ‘slow’). Hidden states evolved according to a Markov chain, and each was associated with distinct probabilities for emitting a high/low intensity stimulus on trial t, conditional on t-1 and t-2 observations.

- The authors investigate two classes of Bayesian learning models (Dirichlet-Categorical models and Hidden Markov Models) that may capture, algorithmically, the inference processes that the somatosensory system approximates during the task.

- The analysis focusses on three model-derived surprise signals, and to what extent they capture variance in the early (< 200 ms) evoked EEG signals ( (1) predictive surprise (i.e. information-theoretic surprise), (2) a confidence-corrected surprise signal, and (3) Bayesian surprise (Kullback-Leibler divergence capturing the belief update following new sensory information).

- They key findings of the paper are (1) variance in evoked EEG responses was best explained by a non-hierarchical learning model (i.e. from the Dirichlet-Categorical class) in conjunction with a leak (forgetting) parameter, (2) EEG mismatch responses from somatosensory cortex covary with model-derived surprise signals, with an early confidence-weighted surprise signal (70ms, S2) followed by a later Bayesian-surprise signal (140 ms, S1). This represents a computationally-plausible sequence whereby an early signal of model inadequacy is followed by (and possibly scales) a belief update.

Strengths –

- The paper represents a rigorous attempt to temporally dissect a prototypical early perceptual surprise signal into distinct computational components (i.e. model inadequacy vs belief updating).

- The focus on somatosensory perceptual learning fills a further gap in the literature, which has largely focussed on neural correlates associated with perceptual surprise signals in the auditory and visual domains (e.g. auditory MMN).

- A further strength is the relatively large sample size (n = 40).

Points for clarification / elaboration

- the Dirichlet-Categorical model, which does not explicitly feature a representation of multiple hidden states and their switches (i.e. is non-hierarchical), was better able to explain the neural data than a Hidden Markov Model (that more accurately corresponds to the true task generative model). The authors comment that perhaps this is evidence that the brain employs simpler (non-hierarchical) perceptual learning models for low level statistical regularity tracking, in the absence of explicit attention to regime switching. I wonder if the authors could comment on how their interpretation interfaces with recent accounts that suggest that even in low-level learning phenomena the brain posits associations between latent causes and observable outcomes (Gershman, Norman and Niv, 2015m Curr. Op. Behav. Sciences).

- I was surprised that the DC model class remained superior to the HMM class even under conditions of perfect integration (no leak), as the leak parameter is precisely what equips the DC model with an ability to be flexible to changes in task statistics. Is it possible that this result (and the superiority of the DC vs HMM in general) is simply due to the emission probabilities associated with the two hidden states being too similar. If so, this limits how generalizable these findings are to task environments with more noticeable transitions between latent states.

- The authors fit the DC model leak parameter separately for each time bin of the evoked response, and found that the optimal parameter corresponding to early periods (where confidence-corrected surprise is encoded) differed from later periods (encoding Bayesian surprise). The authors also suggest that the former signal (CS) may control the latter (BS). Could the authors comment on how the difference in the time-scale of integration between the two signals is likely to affect this interaction.

- The mathematical rigour with which the authors spell out their methods is commendable. However, for very simple points perhaps equations could be omitted to ease readability (e.g. equation 1 which simply reiterates that s(t) is ‘static’).

- Typo Figure 3 legend. Middle is the ‘alternation probability’ mode, not ‘transition probability’ model

Reviewer #2: In this manuscript, the authors investigate Bayesian inference and learning in the somatosensory domain. In particular, they ask which algorithms are used to implement Bayesian belief updating during a somatosensory mismatch paradigm with a roving stimulus. They use both a conventional average-based ERP analysis and single-trial modelling of EEG data from 40 participants performing this task, with both analyses applied both in sensor as well as in source space.

In the model-based analysis, they compare two learning models, both of which derive from a Bayesian model inversion, but using different generative models. Both are ideal Bayesian observer models and no participant-specific parameters are estimated (with the exception of a leak parameter in one of the models). For each model, they consider tracking of four different sequence statistics, and for each combination compute three different trial-wise surprise measures. These measures are subsequently used to predict trial-wise EEG responses, and a hierarchical approach to model selection enables step-wise inference on the generative model, the sequence statistic which is tracked, and the kind of surprise measure.

They conclude that the data are most compatible with a non-hierarchical learning model which estimates transition probabilities between events, and that early signals originating from secondary somatosensory cortex reflect confidence-corrected surprise (a form of puzzlement surprise) and later signals originating from primary somatosensory cortex reflect Bayesian surprise (i.e., model updating).

This is a very well written and interesting report with a novel analysis approach to single-trial EEG data, which enables inference both on the form of the learning model employed and on the nature of the neural surprise signatures from this model at the same time. I would happily support publication of this report in PLOS CB, as it offers both a new analysis framework to compare different models based on observed EEG responses, and has the potential to significantly advance our understanding of the mechanisms underlying somatosensory learning. However, I would challenge the authors to tap this potential further by being more explicit about (1) which learning mechanisms are supported/ruled out by their data, (2) what the different surprise signatures (PS, CS, BS) mean for an implementation of Bayesian learning, and (3) how the model-based results fit together with the MMRs identified in the conventional ERP analysis.

In particular, I would like to see the authors' response to the 4 main points listed below.

Sincerely,

Lilian Weber

Major:

1. First of all, I would challenge the authors with the following claim:

Showing that electrophysiological responses co-vary with specific computational quantities only contributes to a mechanistic understanding of the neuronal computations underlying the learning process, if

a) a concrete implementation of the computations that the quantity is involved in is conceivable (because the results can then be seen as prelim. evidence for such an implementation/neural process), or,

b) the specific quantities or the order of their representation rules out otherwise plausible proposals of the underlying mechanisms (i.e., not all variants of Bayesian inference in the somatosensory system are compatible with the observed pattern of results)

My feeling is that at least one of these is given in the current study, but would love to hear the authors' thoughts on this. I think this would greatly clarify the contribution that the current results make towards a mechanistic understanding of somatosensory learning.

In this context, I would also encourage the authors to carve out the critical difference between the two models they are comparing, to understand why the simpler model fits the data better. In their analysis approach the more complex model, which mimics the data generating process much better than the simpler DC model, is not penalized for complexity (because no subject-specific parameters are fitted). So what is the data feature that the DC-TP1 model captures, but the HMM-TP1 doesn't? E.g., does the HMM-TP1 predict different learning rates (and thus different surprise values) for the two different regimes (the volatile and the stable blocks), which are not supported by the data? Such insight would help to clarify the conclusions we can draw from the data about the learning mechanisms, and relate the results to the literature on whether or not participants adapt their learning rates to the volatility of the environment (e.g., refs 63, 86, 87, Behrens et al. 2007 Nat Neurosci).

2. Secondly, one major claim of the study is that different measures of surprise are represented by EEG signals at different time points and sensors.

I would love to know what the authors think is the functional significance of PS/CS? In particular, in the update equations for the winning (DC) model, PS/CS is never used/computed explicitly. Why would the organism invest the additional energy to compute this (eqs.7,10,12), if it does not have any functional significance in updating beliefs? The authors, in the discussion, hint at a potential role of CS serving to control update rates (p.30, l.666), and interpret their findings as evidence that a higher-level region (S2) represents aspects of confidence, which is used to modulate belief updating on lower levels (S1). In other Bayesian models of inference learning like the HGF, update equations explicitly consider confidence (belief precision) as a driver of learning (update) rates. Such models have been used by our group to understand learning in auditory mismatch paradigms (Stefanics et al. 2018 J Neurosci; Weber, Diaconescu et al. 2020 J Neurosci). Do the authors see their data as compatible with such an account?

3. The authors present two complementary analysis approaches - a conventional average-based ERP analysis, and a single-trial model-based analysis. Both of these drive seemingly independent conclusions about the temporal dynamics of perceptual inference in peristimulus time: the results from the conventional analysis hint at early change detection in S1, then perceptual learning in S1/S2, and later attention-related effects. The results from the single-trial analysis suggest an early representation of CS in S2, and later representation of BS in S1. How do these relate to each other?

I would encourage the authors to address this question, for example by deriving predictions from the different models for MMR effects: (how) does the MMR arise from differences in surprise between trials labeled as standards and those labeled as deviants in the conventional analysis? What predictions do the models make about the effects of train length on surprise? Is the winning model compatible with the experimental observations for the different MMRs?

4. Separate, independent model comparisons are performed for each sensor and peristimulus time bin. (As far as I can tell, the variational inference procedure described in the supplementary section S2 was applied on all of these data points separately.) Can the authors comment on whether this creates a multiple comparison problem and if yes, in how far their analysis deals with this? Does their choice of exceedance probabilities at each step of the hierarchical model comparison, and/or their choice of cluster size thresholds (in time and sensor space) used for detection of significant clusters account for this?

Also, to get an impression of the model fit beyond the relative comparison to other models, can the authors report the % variance explained in the trial-by-trial EEG amplitudes by the winning model?

Minor comments/questions:

Intro:

- p.3,l.66: I find the reference to prediction error confusing here, as (precision-weighted) PE in Bayesian models is often equivalent to model adjustment (Bayesian surprise)

- p.3,l.72 etc.: the introduction of the different surprise measures could be improved. First if all, predictive (Shannon) surprise in practical applications (including here) is computed with reference to subjective beliefs about the probability of events, not the objective frequency. Second, the difference to CS then remains vague, and the mathematical description for CS which is given on p.15 comes very unexpected. Can the authors more clearly state in the introduction what is different in CS from PS (e.g., even if an event is subjectively unlikely (PS), it is not necessarily surprising)?

Methods:

- p.5, l.118: 'oddball-like'?

- p.7: It would be much easier for the reader to first briefly describe the resulting tone sequence and then go into the generative model for it.

- p.8, table1: please provide stimulus stats, e.g. average train length in the two regimes

- p.8, 'Event-related potentials' - given that the GLM already included the parametric regressors for train length, why was this effect further investigated in the significant beta estimates by testing for a linear relationship with train lengths?

- p.11-13: a simple and intuitive description of the DC model learning process might be given, e.g. 'the observer simply counts the observations of each type to determine her best guess of their probability (eq.6), with an exponential forgetting, i.e. discounting observations the further in the past they occurred (eq.9).'

- it seems from figure 6 that catch trials were included for the DC model? If so, why were they modeled for one model, but not the other (HMM)?

- in addition to visualizing the surprise readouts, it would be nice to also visualize the learning process itself, in particular in the DC model (e.g. the evolution of the estimated probability vector alpha over the tone sequence) - a figure for the DC model similar to fig.5 for the HMM.

- p.14, l.327-334: This is not clear. In particular, l. 333 "Thus, the HMM estimates two vectors of emission probabilities corresponding to these events" - which two vectors and which events?

- p.15, figure 5: might be worth mentioning that the 2 states modelled by the SP model do not correspond to the two regimes - the figure might suggest that p(s_t) should track the underlying regimes, while s_t has a different meaning for the SP!

- p.15, l.355: the prior used in CS is not the (flat) prior of the naive observer: CS = KL between the informed prior and the naive posterior.

- p.17: might be worth mentioning that regressors were the same across participants (or if they differed, they only did so because the stimulus differed), and no participant-specific parameters were estimated (except for the optimization of tau)

- p.17 please state the total number of linear regressions run (i.e., number of sensors x number of peristimulus time bins) (i.e., the total number of model comparisons run for 'independent' data points)

- p.18, l.415: and each sensor?

Results:

- p.23, l.490&491: exact p-values and t statistic?

- p.23, l.510-512: please explain this here, so that the reader does not have to refer to the supplementary to understand what is plotted in the scalp topographies. Please state explicitly in the text and the figures what the scalp topographies show, and what this parameter means.

- figure 12 and figure S3: please state explicitly which steps resulted in data reduction (i.e., a selection of EEG sensors and time points for which a meaningful model comparison results could be retrieved, to be included in the comparison at the next step)

- figure 13: please state the unit for the half-life (observations?)

Discussion:

- p.26, l.548-550: the interpretation, especially of the P300 effect as an attention-allocating process, comes somewhat ad-hoc, because it hasn't been motivated before. It is discussed later again (p.27, l.580-6), but only afterwards (p.28) are the aspects of the current results mentioned which support this interpretation (i.e., linear dependence of the P300 to deviants on train length). If this is the main finding that the authors base their attentional interpretation on, it should be mentioned earlier.

- p.29, l.622-625: Not sure I understand this conclusion. The winning model did not learn about the different regimes in any way, so neither explicit nor implicit learning of the regimes are supported.

- p.29, l.630: one of the cited studies (ref. 86) employed a very different form of hierarchy without an explicit representation of change points. This model (the HGF) is actually more similar to the non-hierarchical DC model used here, except that the leakiness (learning rate) is a function of a subjective estimate of volatility (i.e., continuous rate of change).

Reviewer #3: In this paper, Gijsen, Grundei and colleagues present a somatosensory mismatch EEG study on 40 participants. Using a roving paradigm of electrical stimuli to the wrist with fixed stimulus probabilities but two different levels of transition probabilities, they examine mismatch responses. The authors first demonstrate somatosensory ERPs and mismatch responses in expected temporal windows. They then use SPM to identify three spatiotemporal clusters with significant differences between standard and deviant trials. The sources of these three “components” are identified using SPM, first as distributed sources and second as equivalent current dipoles. Finally, the authors examine trial by trial responses. For this they model the trial by trial variations of every single point in the ERP with a series of models that are built along 3 axis (1. Dirichlet categorical vs. hidden markov model; 2. Whether inference was done on stimulus probability, alternation probability, transition probability (1st order) or transition probability (2nd order); 3. Three different surprise readouts: predictive surprise, bayesian surprise and confidence-corrected surprise). The authors then invoke a hierarchical family/model comparison to come to the conclusion that “a non-hierarchical Bayesian learner performing inference on transition probabilities” explains their data best with different forms of surprise in different time windows and different topographies. In addition, the authors show that forgetting over a time window of roughly 50 trials provides the best fit to the data.

This is a very nice study. The manuscript is clearly written and nicely combines robust experimental work (40 subjects) with mathematical modelling. In an improvement of standard model based EEG/fMRI studies, the authors also formulate the regression model of the predicted time courses onto the data as a Bayesian regression, which allows them to do Bayesian model comparison. I think this paper fits nicely the scope of PlosCB. I have a couple of major points which I think the authors would need to address before publication. All of these major points have to do with the model selection and Bayesian inference part. See more details below.

Direct comments to the authors:

Major points:

Hierarchical model comparison approach: You have chosen to act against the “dilution of evidence” across many models by invoking a hierarchical model comparison scheme based on exceedance probabilities in a series of hierarchical comparisons. While there are other examples of such a hierarchical approach in the literature, this is, to my knowledge, not standard in family comparison of models, and I am not aware of any paper that suggests that this procedure is correct for selecting the best model. Every model or family comparison is conditional on the model space that you put in. In the extreme case, your final set of three models that you compare might not even include the best of all models. I would recommend running a model comparison over all models and running three family comparisons where you arrange your families to compare models along the three dimensions on your model space. Even if model comparison turns out to be inconclusive, this is an important information for the reader and the family comparisons should allow you to make some general statements about the different dimensions of your model space, which are of interest. In conclusion, I think that using the hierarchical scheme, you cannot safely conclude that “EEG signals were best described using a non-hierarchical Bayesian learner performing transition probability inference.” But, maybe the search for a single best model is not even the most important goal here if you can make more robust and solid statements about other dimensions, e.g. whether an HMM or DC is better, or which kind of Surprise explains the data best at what time point, irrespective of the precise formulation of the other aspects of the model.

Exceedance probabilities: You use exceedance probabilities for all comparisons. These are known to be inflated and should whenever possible be replaced by protected exceedance probabilities (Rigoux et al, Neuroimage, 2014, doi: 10.1016/j.neuroimage.2013.08.065). I think it would be good if you showed plots of the expected probabilities for all comparisons if you cannot use protected exceedance probabilities which unfortunately are not available for family comparisons. Seeing the expected probabilities will give the reader an idea of the probabilities of individual models and families.

Fitting of tau and model evidence correction: In order to correct for the fitting of tau (the forgetting in the DC models), you subtract “the degree to which tau optimization on average inflated model evidences”. First, I do not fully understand the procedure. Average over subjects, over voxel-timepoints? Second, I am not sure this heuristic properly accounts for the additional complexity introduced by tau. Do you have a reference that shows that this heuristic properly controls for complexity? You might be correcting too little or even too much, in which case, your results would become even clearer. In favor of your selection of DC as the winning model class you state in the discussion that the HMM did never win, when tau=0. Does that mean that the DC still clearly won in all these cases? I think you should show the same map as in Figure 12A also for the case of tau=0. This would help to understand the impact of fitting tau. Ideally, the fitting of tau (including defining a prior) should be part of the model inversion, but this might be a larger effort going beyond the scope of this paper. However, I think you should mention this option in the discussion.

Conclusion of Bayesian learning: You conclude that “early somatosensory cortex seems to reflect Bayesian perceptual learning” (lines 733/734). From your analysis, it is difficult to make a statement about the Bayesian part. All learning models that you tested are Bayesian in nature (except for the null model), hence it could well be that a non-Bayesian model could also provide a good explanation of the data. We simply do not know.

Inconsistencies in hierarchical scheme: There are a couple of questions to your hierarchical scheme. These are however only relevant, if you would like to stick to it. I just mention them here, and I think you would have to answer them convincingly, if you stick to this scheme.

1.) Why are the thresholds changing for every level?

2.) What is the rational for splitting up the comparison over TP1, TP2, SP and AP into two? I think this should be one single model comparison. (In fact, this leads to a misinterpretation of results when you say “Our results show that the TP model family clearly outperformed the SP and AP families.” What you show is that TP1 outperforms these other families.) Why are you reevaluating in places/at timepoints where TP1 does not win? This deviates from the general strategy.

3.) Why did you choose this exact order of hierarchy? What would a different ordering yield?

4.) Even if you stick to the hierarchical scheme, which I do not recommend, I think you would have to show the expected model probabilities for all models and family comparisons. The reader should be able to appreciate that the final decision for a single model, although it might be clear in the final step, is only performed within a probably small fraction of the entire mass of your model space. It is probably not feasible to show this for all voxel-timepoints, but you could select a couple of representative examples.

5.) How can you assure that your statements about the best model hold?

Minor points:

Multiple comparison for Bayesian Model Selection: Doing model or family comparison for every single voxel-timepoint means that you are conducting many model comparison tests. I am not sure there is a solution for this problem, but it might be worth mentioning this. I do not think this invalidates any findings at particular levels, but it might be good to remind the reader that the voxel-timepoints with preference for a particular family are just few of many that were tested.

Line 189: How was train length entered in the GLM? As a parametric modulator, or as several modulators each coding for one length?

Line 255: I think there is a typo in the right hand side of the equation. One of the j indexing s_t and s_t-1 should be an i.

Fig. 5: The x-axis label is probably trial number and not time in ms.

Fig 8: Reference to panel E is missing in caption.

Fig 9: Please remind the reader of the coloring of deviants and standards (bottom row). I assume this is the same coloring as in Figure 1.

Fig. 10: Are the values for rS2 and lS2 correct. Shouldn’t the “Moment Posterior” be symmetric as well?

Congratulations on this nice study.

Jakob Heinzle

**Have all data underlying the figures and results presented in the manuscript been provided?**

Reviewer #1: Yes

Reviewer #2: Yes

Reviewer #3: Yes

PLOS authors have the option to publish the peer review history of their article (what does this mean?). If published, this will include your full peer review and any attached files.

Reviewer #1: No

Reviewer #2: **Yes: **Lilian Weber

Reviewer #3: **Yes: **Jakob Heinzle
---

## [Decision Letter · Decision Letter 1]

16 Oct 2020

Dear Dr Gijsen,

Thank you very much for revising and re-submitting your manuscript "Neural surprise in somatosensory Bayesian learning" for consideration at PLOS Computational Biology.

As with all papers reviewed by the journal, your manuscript was reviewed by members of the editorial board and by several independent reviewers. In light of the reviews (below this email), we would like to invite the resubmission of a significantly-revised version that takes into account the reviewers' comments.

The general assessment of the referees is still favorable both with respect to the manuscript itself and the efforts taken during the revision. However, two reviewers raise important issues that have not been sufficiently addressed in those revisions.

In particular, both reviewers raise concerns about several aspects regarding the nature of the model comparisons and thresholding as well as dealing with multiple comparison issues. These issues mainly relate to the statistical reporting and interpretation of some of the claims made in the manuscript. Further, one reviewer suggests to investigate predictions from the different models for ERP differences between standard and deviant trials in more detail. Please see the detailed comments below.

We cannot make any decision about publication until we have seen the revised manuscript and your response to the reviewers' comments. Your revised manuscript is also likely to be sent to reviewers for further evaluation.

Sincerely,

Philipp Schwartenbeck

Guest Editor

PLOS Computational Biology

Samuel Gershman

Deputy Editor

PLOS Computational Biology

Reviewer's Responses to Questions

**Comments to the Authors:**

Reviewer #1: Thank you for asking me to re-review this manuscript. I would like to thank the authors for engaging with my comments, which primarily concerned the interpretation of the findings. I am satisfied both by their responses and additions to the Discussion section. I believe the manuscript has been greatly improved.

Reviewer #2: The authors have put some work in addressing my concerns. I particularly found the new Figure 5 very insightful. However, I am not fully convinced by all of their responses. In particular, this concerns:

a) the relationship between the conventional ERP analysis and the model-based single-trial analysis

b) their statistical thresholds and analysis choices.

These points would need to be addressed before I could support publication.

Sincerely,

Lilian Weber

Comments to the authors:

Thank you for comprehensive replies to my concerns and the considerable effort you put into this revision. I particularly appreciated Figure 5, and some of the clarifications you provided in your responses regarding your overall hypotheses and the scope of your approach.

However, I was not convinced by some of your replies. I list these below with comments.

a) conventional ERP results and single-trial model-based analysis

In my previous comment, I wrote:

"I would encourage the authors to address this question, for example by deriving predictions from the different models for MMR effects: (how) does the MMR arise from differences in surprise between trials labeled as standards and those labeled as deviants in the conventional analysis? What predictions do the models make about the effects of train length on surprise? Is the winning model compatible with the experimental observations for the different MMRs?"

I appreciated your addition to the Discussion about the potential relationship between ERP components and the EEG correlates of your surprise measures, which I found offered a very sensible interpretation. However, I still think you could make much more specific statements by looking at what the different models would predict in terms of a standards vs deviants contrast. Importantly, this to me does not seem to require a disproportionate effort: you only have to apply the conventional trial definition to your model-based surprise readouts. After all, you motivate your

study with the question of which mechanisms underlie the classically observed mismatch signals (l.35-38).

Figure 5 was already helpful in understanding the specific predictions that the different models make for single-trial ERP responses, which could explain their differential performance in predicting the EEG amplitudes. For example, CS in the DC model predicts these slow drifts within trains of stimuli, where variance (between standards and deviants) decreases, but the overall mean surprise increases. Using these model-based single-trial surprise measures you can easily derive MMR predictions by averaging surprise to standards versus deviants.

For example, this could illustrate your point: "This counteracting effect of belief commitment and the surprise terms can lead to independence of CS and train length when responses are averaged", or you could show which surprise measures would predict the lack of difference between MMRs in the stable versus the volatile regime.

The traditional definition of standards and deviants is a heuristic for what is surprising or predictable, and implicitly suggests a model of how observers perceive the sequence: e.g., repetitions are more likely than transitions. Or: observers track the stimulus probability for every trial, and settle on a higher probability of the repeated stimulus towards the end of a train, then update at the onset of the new train. Your single-trial analysis uses more information (i.e., all trials) from the data to make a more precise statement about underlying mechanisms, but from your current results it remains unclear if the classically observed MMRs correspond at all to differences in surprise between standard and deviant trials as quantified by your winning model.

b) analysis choices

- multiple comparisons:

You write: "In Bayesian model comparison there is no conventional way to correct for multiple comparisons and it has been established that Bayesian methods provide inherent adjustments of sensitivity and specificity to deal with false positive rates (Friston 2002, Neuroimage, doi:10.1006/nimg.2002.109 and Friston 2002, Neuroimage, doi:10.1006/nimg.2002.109)."

I cannot agree with this point and I don't see how the cited papers relate to your analysis. As far as I can see, these deal with hierarchical Bayesian models (parametric empirical Bayes), whereas you perfom separate, independent model comparisons per voxel (sensor and time point). I'm happy to be corrected here.

- catch trials:

I appreciate your reanalysis and including it as a supplementary figure. It does seem to me that when excluding catch trials for the DC model, the evidence for the DC model compared to the HMM is significantly reduced (Fig. S7B vs S7A). Given that the comparison without the catch trials is the fairer one, I think this deserves mentioning in the main text.

- exceedance probability thresholds in the step-wise comparison approach

The different model comparisons (DC vs HMM, TP1 vs TP2, etc.) seem to be orthogonal to each other. Therefore, I don't understand why the thresholds should decrease over the successive steps. The fact that the voxels have been thresholded before does not seem relevant

if the comparisons are orthogonal?

- comparisons between different surprise readouts

Judging from Fig. 5, I would expect the different surprise measures to be hardly distinguishable, especially BS and PS in the DC model. Indeed, when using protected exceedance probabilities, the evidence for one surprise measure over another seems to be weak at best (Fig. S8). Given that the 'alternative statistics' (expected probabilities and protected exceedance probabilities) are actually the more robust statistics, the data do not seem to provide strong support in favour of one or another surprise measure. (The fact that the results with a lower threshold, i.e., the exceedance probabilities, look similar to the ones with a higher significance threshold does not mean that the exceedance probabilities are not inflated.) I believe it would be appropriate to tone down your conclusions about different surprise measures reflected in your data.

Other issues:

- You write in the abstract:

"As such, this dissociation indicates that early surprise signals may control subsequent model update rates."

I don't see how this is indicated by your data/results. It is a plausible interpretation.

- In your response letter:

"That is to say, the currently tested models do not provide a plausible manner by which the brain acquires the estimated transition probabilities and subsequent surprise quantities. Rather, we view our model comparison as a methodology to infer on qualities that a future successful neural algorithm is likely to exhibit (e.g. using estimated transition probabilities to compute an early puzzlement surprise signal scaled by

confidence)."

I appreciate this perspective and think it would be worthwhile sharing this with the reader as well.

Reviewer #3: I would like to thank the authors for their explanations and additional analysis, which clearly help understand the results and also support many of their conclusions. However, I woudl encourage them to include more important information abuout statistics in the main manuscript (for example the expected posterior probabilities and the protected exceedance probabilities are provide to the supplement). Finally, there still remain some open questions regarding the hierarchical scheme, model comparison and thresholds which I think were not answered sufficiently, yet.

Major:

Hierarchical scheme: First of all, thank you for this extensive reply. I used the term hierarchical in order to reflect how you named it in the original submission it was not my intention to imply that this was a truly hierarchical Bayesian scheme. You have decided to stick with the “hierarchical model selection” scheme. I still think, this is not think a standard in DCM research. There might be some papers which have used family model comparison in this way, but to my best knowledge there is no theoretical or methods paper suggesting this. I do not recall the original paper by Penny and colleagues does promote family comparison for this purpose either. If you are aware of one or several citations that advocate such a hierarchical selection approach and evaluate it critically, I would be more than happy to know it, and you should cite them in your paper. Having said this, your approach is reminiscent of using orthogonal contrasts in a factorial design to reduce the search space and if one would assume that model selections are orthogonal (which I think one can) selecting certain time points based on one comparison and then restricting the rest of the analysis to those should be fine.

However, the combined reduction of search volume and model space is to my best knowledge a novel approach. Hence, you should discuss it critically.

I have some additional comments to your answers to my requests for clarification on the hierarchical scheme. None of these points is new. They are all related to your answers to the previous comments.

Regarding thresholds you say: “We allowed for lower thresholds for these second and third analysis steps on remaining data given a threshold had already been applied.” I do not understand this rational. Why should a threshold on a lower level be less stringent than on a higher level of your selection? If these are orthogonal questions, which is how you treat them in your comparison, all comparisons should have the same threshold, I would think. It would be much more convincing if you used the same exceedance probability threshold for all levels (for example 0.95, which would roughly correspond to a p-value of 0.05 (see also my comment below)). If you stick to the thresholds you selected, I think you should remove the above sentence and make clear that this was an arbitrary choice.

Regarding statistics and thresholding: I think you should mention expected posterior probabilities and protected exceedance probabilities directly in the manuscript. The reader should have an idea of the size of the effect from reading the main text. While there is not a one to one mapping from <r> to exceedance probability or to protected exceedance probabilities, you could say something like: “For the individual levels we thresholded at phi = 0.99 which roughly corresponded to <r> = 0.7??, phi = 0.95 (<r> = ??? etc.), and phi = 0.9 (<r> = ???, protected exceedance probability = ???) and phi = 0.7 (<r> = ???, protected exceedance probability = ).” You could then refer to the supplementary figure to illustrate all time points and electrodes.

I think it would be best, if you used protected exceedance probabilities for the final level. Why would you apply a measure that we know is inflated if you have a robust alternative? From the figure you show in the supplement it seems that there is not much information in the data to distinguish the surprise models. Rigoux and colleagues suggest that one minus the protected exceedance probability could be used similar to a p-value. Hence, an unprotected exceedance probability threshold of 0.9 is low (this is comparable to a p-value of 0.1, but still not considering the null hypothesis of all models being equally likely) and a threshold of 0.7 (p<0.3) seems extremely low. I think this should be made clear to the reader and it might be more correct to not call these findings significant. I do not think it is critical that there is strong evidence for one particular model, but as it stands I fear you tend to interpret relatively little evidence as a strong finding. Finally, the fact that more stringent statistics like protected exceedance probability reveal a similar pattern although at lower values should not be taken as a confirmation of the inflated values of the exceedance probability. This is what you seem to suggest: “Despite these statistics being diminished, they yield highly similar conclusions, suggesting the results are not solely due to exceedance probability inflation.” Of course, the conclusions depend on thresholds. Overoptimistic values of exceedance probability will not change the overall pattern of the maps, but could potentially make us overoptimistic about conclusions. For example, the protected exceedance probability maps suggest that there does not seem to be much evidence in favor of any of the surprise models.

Finally, I think the family comparison where you stick to the factorial design is indeed convincing. And supports your findings within the DC group: TP1 is the clear winning family. Again, as in the hierarchical selection procedure, there seems to be rather little evidence in favor of any particular surprise model. I have one small question for clarification. Did you really not reduce data (the time points and electrodes) in this comparison, or did you just not reduce the model space?

From all the points raised above, I would conclude the following. You can provide robust evidence that the DC is favored over NULL and HMM. In addition, it is still quite clear that TP1 outperforms (TP2, AP and SP) models. For the different surprise models, the evidence seems to get rather week. One could maybe talk about a tendency of some models to win.

Multiple comparison: I think this needs more discussion. I am not aware of work that says that performing thousands of independent Bayesian analyses cannot result in an issue of multiple tests. I am more than happy to be corrected on this. I interpret the statement in the Friston 2002 paper you cite differently. Their setting differs in two important aspects from your analysis. First, it is about parameter estimates. Second, and more importantly, the comment about solving the multiple comparison problem is made in the context of a hierarchical model (PEB) where the higher level serves to set the prior according to the distribution over all other voxels (empirical Bayes). It is this step that “provides” the correction. I do not think the situation is the same in your setting. I think one needs to acknowledge the fact that this is an unsolved problem and discuss it accordingly.

Minor:

Fitting of tau: If there is a citation where your heuristic to correct for the fitting of tau is suggested, you should cite it here. Otherwise, please state that it is a heuristic that somehow punishes for the additional fitting but that one would have to do include tau as an additional parameter in the model fitting to do proper model comparison including tau.

In supplementary figure 4, it looks as if a threshold of 0.9 was used to go from level 2 to level 3. However, in the manuscript and the corresponding figure you write 0.95. Please correct the one that is wrong.

The simulation you perform is illustrative but difficult to assess without knowing more details. In particular, it would be interesting to know how you simulated different settings of families. Did you sample from a Dirichlet distribution using the posterior of your analysis or did you assume that all 40 subjects have the same model. The latter would probably be quite an extreme case. In any case, I am not so sure a simulation can proof that the method is correct. But, it can already show some limitations, which seem to occur for certain models even in this ideal scenario.

In summary, I still consider this a highly valuable contribution for PLOS CB, but I would think that the issues raised above should be considered.

Jakob Heinzle</r></r></r></r></r>

**Have all data underlying the figures and results presented in the manuscript been provided?**

Reviewer #1: **No: **n/a

Reviewer #2: **No: **Will be made available upon acceptance as indicated in the "Data Availability" section.

Reviewer #3: Yes

PLOS authors have the option to publish the peer review history of their article (what does this mean?). If published, this will include your full peer review and any attached files.

Reviewer #1: No

Reviewer #2: **Yes: **Lilian Aline Weber

Reviewer #3: **Yes: **Jakob Heinzle
---

## [Decision Letter · Decision Letter 2]

18 Dec 2020

Dear Mr Gijsen,

We are pleased to inform you that your manuscript 'Neural surprise in somatosensory Bayesian learning' has been provisionally accepted for publication in PLOS Computational Biology.

Best regards,

Philipp Schwartenbeck

Guest Editor

PLOS Computational Biology

Samuel Gershman

Deputy Editor

PLOS Computational Biology

Reviewer's Responses to Questions

**Comments to the Authors:**

Reviewer #2: I want to thank the authors for their efforts in addressing my remaining concerns. I think the results from the ERP simulations are very interesting and help connect the model-based and the conventional perspective on MMR. I believe the methods and results are now presented in a transparent way and that the study makes a very valuable contribution to our understanding of the neural signatures of somatosensory learning. I enjoyed reviewing this interesting manuscript and look forward to seeing the paper in PLOS CB!

Sincerely,

Lilian Weber

Reviewer #3: This revision is again much improved. The authors have dealt with all the issues I had raised in a satisfactory way by either solving the problem or discussing it adequately. I recommend accepting the manuscript.

I have one minor comment which however can be easily corrected (maybe even in the proofing stage).

Minor: I have not come across the notion that exceedance probabilities are a measure of effect size, and to my understanding, they are not. I would suggest you remove the term “effect size” from the manuscript and simply say exceedance probabilities (or probability). Effect size is associated with a specific meaning and it might confuse readers if you call exceedance probabilites effect sizes.

Once again, congratulation on this work

Jakob Heinzle

**Have all data underlying the figures and results presented in the manuscript been provided?**

Reviewer #2: **No: **Data will be made available upon acceptance.

Reviewer #3: None

PLOS authors have the option to publish the peer review history of their article (what does this mean?). If published, this will include your full peer review and any attached files.

Reviewer #2: **Yes: **Lilian Aline Weber

Reviewer #3: **Yes: **Jakob Heinzle

---

## [Editor Report · Acceptance letter]

23 Jan 2021

PCOMPBIOL-D-20-01012R2 

Neural surprise in somatosensory Bayesian learning

Dear Dr Gijsen,

I am pleased to inform you that your manuscript has been formally accepted for publication in PLOS Computational Biology. Your manuscript is now with our production department and you will be notified of the publication date in due course.

With kind regards,

Alice Ellingham
